

# A comparison of two chemistry and aerosol schemes on the regional scale and resulting impact on radiative properties and warm and cold aerosol-cloud interactions

Franziska Glassmeier[1,2], Anna Possner[1,3], Bernhard Vogel[4], Heike Vogel[4], and Ulrike Lohmann[1]

[1]Institute for Atmospheric and Climate Science, ETH Zurich, Zurich, Switzerland
[2]*current affiliation:* Chemical Sciences Division, NOAA Earth System Research Laboratory, Boulder, USA
[3]*current affiliation:* Department of Global Ecology, Carnegie Institution for Science, Stanford, USA
[4]Institut für Meteorologie und Klimaforschung, Karlsruhe Institute of Technology (KIT), Karlsruhe, Germany

*Correspondence to:* Franziska Glassmeier (franziska.glassmeier@noaa.gov)

**Abstract.** The complexity of the atmospheric aerosol causes large uncertainties in its parameterization in atmospheric models. In a process-based comparison of two aerosol and chemistry schemes within the regional atmospheric modeling framework COSMO-ART, we identify key sensitivities of aerosol parameterizations. We consider the aerosol module MADE in combination with full gas-phase chemistry and the aerosol module M7 in combination with a constant-oxidant-field-based sulfur cycle. For a Saharan dust outbreak reaching Europe, modeled aerosol populations are more sensitive to structural differences between the schemes, in particular the consideration of aqueous-phase sulfate production, the selection of aerosol species and modes and modal composition, than to parametric choices like modal standard deviation and the parameterization of aerosol dynamics. The same observation applies to aerosol optical depth (AOD) and the concentrations of cloud condensation nuclei (CCN). Differences in the concentrations of ice-nucleating particles (INP) are masked by uncertainties between two ice-nucleation parameterizations and their coupling to the aerosol scheme. Differences in cloud droplet and ice crystal number concentrations are buffered by cloud microphysics as we show in a susceptibility analysis.

## 1 Introduction

Atmospheric aerosol poses the most uncertain factor in quantifying the anthropogenic forcing of the climate system (Myhre et al., 2013). This uncertainty is rooted in the complexity of aerosol characteristics and processes: Aerosol particles feature many microscopic degrees of freedom like their chemical composition, mixing state or shape, and interact with several atmospheric components like atmospheric chemistry, the planetary surface as source of primary emissions, radiation by scattering and absorption and the hydrological cycle via aerosol-cloud interactions (Lohmann et al., 2016). Given their microscopic scale, all these processes and characteristics have to be parameterized to be represented in atmospheric models.

Approaches to represent aerosol particles in atmospheric models employ discrete (binned) or continuous (modal) distributions of particle sizes (Jacobson, 2005). They consider different selections of chemical species like sea salt, dust, sulfate, nitrate and classes of organics, e. g. soot, primary or secondary organic aerosol, that are grouped in internally and/or externally mixed



particle classes. The parameterizations of aerosol microphysical processes like gas-to-particle conversion, coagulation and dry and wet deposition depend on these structural aerosol characteristics (e. g. Vignati et al. (2004); Vogel et al. (2009)).

Modeled aerosol particles can be coupled to an atmospheric host model to different degrees: Atmospheric chemistry can be considered from simplified sulfur cycles using climatological oxidant fields (e. g. Zubler et al. (2011)) to full chemistry including aqueous-phase reactions (e. g. Knote and Brunner (2013)). Primary aerosol emissions may be prescribed from inventories or modeled online taking into account surface conditions (e. g. Vignati et al. (2004); Vogel et al. (2009)). Aerosols can also be coupled to radiation via their absorbing and scattering properties and to cloud formation by their ability to serve as cloud condensation nuclei (CCN) or ice nucleating particles (INP) (Lohmann et al., 2016). The latter aerosol-cloud interactions are the largest contributor the the uncertainty of the aerosol forcing (Myhre et al., 2013). Clearly, the challenge lies in choosing the right degree of complexity for a given task, e. g. air quality or climate projections. An informed choice requires an understanding of key processes and sensitivities of aerosol parameterizations.

While aerosol microphysics take place on the microscale, aerosols can be transported globally (Lohmann et al., 2016). Regional atmospheric models are valuable tools to increase our process understanding because they compromise between process representation that is improved at higher spatial resolutions and larger-scale transport patterns (e. g. Possner et al. (2015), Rieger et al. (2014); Athanasopoulou et al. (2013); Knote and Brunner (2013); Bangert et al. (2012); Fountoukis et al. (2011); Zubler et al. (2011). Nevertheless, our current understanding of aerosols remains insufficient (Myhre et al., 2013). While for air quality applications in general and case studies in particular, the uncertainties in aerosol representation can be somewhat controlled by tuning the parameterization to match observations, reducing the uncertainty of climate predictions depends on improving our understanding of key sensitivities of aerosol parameterizations (Lee et al., 2016).

Multi-model intercomparisons and sensitivity studies using a single model are complementary approaches to assess uncertainties of aerosol parameterizations: Intercomparisons compare different representations of aerosol characterizations, process parameterizations and parameter choices in a statistical fashion. Observed differences are judged in comparison to observational data and can usually not be attributed to specific processes or characteristics and their implementation. The AQMEII initiative is an example of a statistical intercomparison and evaluation of multiple regional aerosol and chemistry transport models and reports large variability between different models that seems related to aerosol deposition but could not be explained on the process level (Solazzo, 2012). On the global scale and with a focus on climate applications, the AeroCom multi-model intercomparison initiative likewise reports large model diversity and concludes from observational bias that emissions and gas-to-particle conversion are insufficiently understood (Mann et al., 2014). Differences in model performance could not be attributed to specific process parameterizations in most cases, however.

Numerical sensitivity studies test the effect of changing a certain parameter or the description of a specific process or aerosol characteristic on the variables of interest and can help to explain model variability. A sensitivity study of model performance to updated process representations, for example, allows Zhang et al. (2012) to attribute an improvement in modeled aerosol water content in comparison to the AeroCom multi-model mean to a $\kappa$-Köhler approach to water uptake. Lee et al. (2012) assess the parametric uncertainty of simulated cloud condensation nuclei (CCN) concentrations using an emulator technique that reveals





the importance of interactions between different parameters and thus highlights the importance of comparing specific sets of parameters and parameterizations rather than varying them one at a time.

This study might be considered a hybrid between the model comparison and sensitivity studies discussed above and naturally takes into account combinations of parameters and parameterization approaches: We will present a detailed comparison of two
different modal aerosol schemes, one developed by the climate community and one that emerged from air-quality and weather predication applications, that are embedded into the same regional atmospheric model. This study intends to highlight key sensitivities to be considered when designing or choosing a modal aerosol scheme. It does not aim to identify the 'better' of the two schemes, which will depend on the specific application. Our analysis comprises targeted sensitivity studies that require an adapted setup of the two aerosol schemes as well as a model comparison of both schemes in their default setups. For the latter,
we additionally discuss resulting impacts on the radiative aerosol properties and implications for warm and cold aerosol-cloud interactions.

The rest of the paper is organized as follows: Detailed model descriptions are given in Section 2. Section 3 describes the different model setups that our analysis is based on. Section 4 compares adapted version of both aerosol schemes in a sensitivity study, while Section 5 is concerned with the differences between the two schemes in their default setups as well as
aerosol optical properties and aerosol-cloud interactions. We summarize and conclude in Section 6. An earlier version of this study constitutes a chapter of the doctoral thesis of Franziska Glassmeier (Glassmeier, 2016).

## 2   Model descriptions: COSMO-ART and COSMO-ART-M7

We employ the atmospheric aerosol and chemistry modeling framework COSMO-ART (Vogel et al., 2009), which is based on the regional atmospheric model COSMO (**Co**nsortium for **S**mall-Scale **Mo**delling, www.cosmo-model.org). The ART
(**A**ersosol and **R**eactive **T**race gases) extension of COSMO features online-coupled gas-phase chemistry and the modal 2-moment aerosol scheme MADE as well as aerosol-radiation and aerosol-cloud interactions. COSMO-ART has a tradition of air-quality modeling and has been extended to investigate the role of interactive aerosol in weather prediction (e. g. Bangert et al. (2012), Rieger et al. (2014)).

We compare this standard version of COSMO-ART to a new assembled model version called COSMO-ART-M7. This new
version integrates the modal 2-moment aerosol module M7 (Vignati et al., 2004; Stier et al., 2005) and the computationally efficient sulfur chemistry of Feichter et al. (1996) as an alternative to the full chemistry and MADE into the COSMO-ART framework. The efficient chemistry is implemented using the code-generator KPP (Damian et al., 2002) that is available within COSMO-ART. The implementation of the aqueous-phase chemistry relies on the reaction rate implementation from GEOS-CHEM (map.nasa.gov/GEOS_CHEM_f90toHTML/). Our implementation of the Feichter sulfur cycle is coupled to the
updated version of the M7 aerosol microphysics as implemented in the global climate model ECHAM-HAM2.2 (Zhang et al., 2012). Primary emissions, dry and wet deposition as well as aerosol-cloud interactions from COSMO-ART are adapted to M7 aerosol modes. The implementation of aerosol-optical properties is M7-specific and described in Zubler et al. (2011).





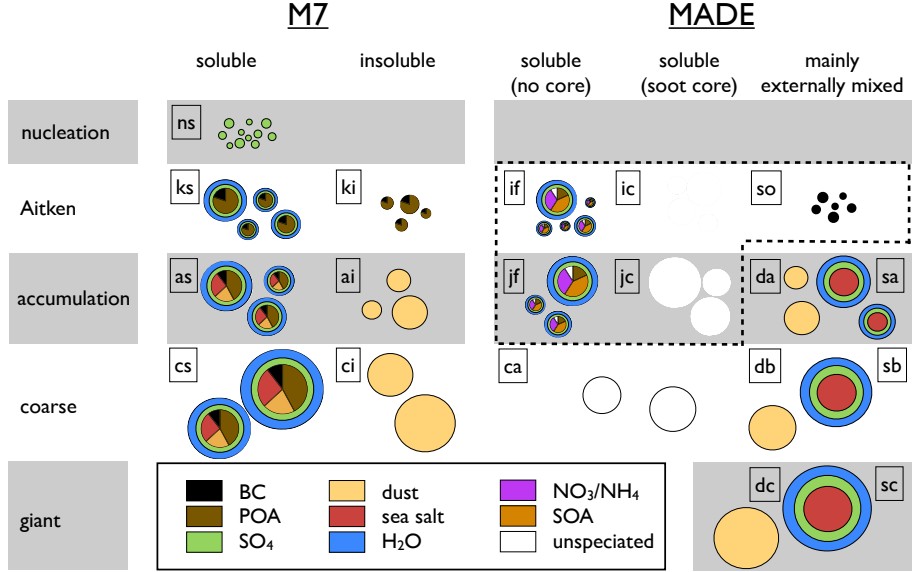

**Figure 1. Comparison of chemical composition of aerosol modes for MADE and M7.** The dashed line indicates modes that are considered for inter- and intra-modal coagulation in MADE. For M7, all modes participate in coagulation or intra-modal transfer by coagulation. The size and standard deviation of modes can be determined from Tab. 1 based on the two-letter abbreviations stated at the upper left of each mode.

The M7 module has been developed for climate applications in global models. COSMO-ART-M7 can be considered an updated version of COSMO-M7 (Zubler et al., 2011): Next to the current versions of COSMO and M7, COSMO-ART-M7 profits from the state-of-the-art droplet activation and ice nucleation parameterizations of COSMO-ART. In contrast to COSMO-M7, COSMO-ART especially includes aerosol-cloud interactions in cirrus clouds. The remainder of this section provides details on

5   the parameterizations and adaptations.

## 2.1   Aerosol

The aerosol module MADE of COSMO-ART represents atmospheric aerosol by 12 coated and uncoated log-normal modes in the Aitken, accumulation, coarse and giant size range. For the composition of aerosol particles, 13 chemical species are considered: dust (DU), sea salt (SS), sulfate ($SO_4$), nitrate ($NO_3$), ammonium ($NH_4$), black carbon/soot (BC), primary organic

10   carbon (POA), 4 volatility classes for secondary organics (SOA) (Athanasopoulou et al., 2013) and unspeciated PM2.5 and PM10 of anthropogenic origin. Based on two-letter abbreviations for each of the 12 modes, Figure 1 and Table 1 summarize the chemical composition, modal standard deviations and initial radii.





**Table 1. Comparison of modal parameters for MADE and M7 modes.** Each mode is identified by a two-letter abbreviation (italic font), which allows to identify its chemical composition in Figure 1. Modal standard deviation is denoted by $\sigma$. Modal count median radii $r$ refer to initial and emission radii for MADE. In contrast to MADE, M7 features a mode repartitioning ensuring that the radii of M7 modes are restricted to the indicated ranges. Mode reorganization in MADE is limited to ensuring that the radii of Aitken modes remain smaller than that of accumulation modes. MADE and M7 modes are grouped to show correspondence.

| mode | nucleation | Aitken | | | accumulation | | | | coarse | | | giant | |
|---|---|---|---|---|---|---|---|---|---|---|---|---|---|
| MADE | - | *if* | *ic* | *so* | *jf* | *jc* | *sa* | *da* | *ca* | *sb* | *db* | *sc* | *dc* |
| $\sigma$ | - | 1.7 | 2.0 | 1.4 | 1.7 | 2 | 1.9 | 1.7 | 2.5 | 2 | 1.6 | 1.7 | 1.5 |
| $r/\,\mu m$ | - | 0.005 | 0.04 | 0.04 | 0.035 | 0.04 | 0.1 | 0.75 | 0.5 | 0.5 | 3.35 | 6.0 | 7.1 |
| M7 | *ns* | *ks* | | *ki* | *as* | | | *ai* | *cs* | | *ci* | - | |
| $\sigma$ | 1.59 | 1.59 | | 1.59 | 1.59 | | | 1.59 | 2.0 | | 2.0 | - | |
| $r/\,\mu m$ | <0.005 | 0.005 – 0.05 | | | 0.05–0.5 | | | | >0.5 | | | - | |

Inter- and intra-modal coagulation is considered for anthropogenic Aitken and accumulation modes (modes labeled if, ic, so, jf and jc in Table 1) but omitted for the sea salt (sa, sb, sc) and dust modes (da, db, dc) and the PM10 mode (ca) as indicated by the dashed line in Figure 1. Sources of MADE aerosols include primary emissions of SS, DU, POA, BC, PM2.5 and PM10 and gas-to-particle conversion of $SO_4$, $NO_3$, $NH_4$ and SOA. Emissions of SS (Lundgren, 2012) and DU (Vogel et al., 2006)

are calculated online based on wind speed. Primary anthropogenic aerosols are based on emissions inventories. Emitted BC is assigned to the pure soot mode (so) and POA is distributed to the Aitken (if) and accumulation mode (jf) without soot core. The POA partitioning follows the emission pre-processor described in Knote (2012). Emissions are assumed to follow the initial modal size distributions summarized in Tab. 1. SOA, $NO_3$ and $NH_4$ condense onto existing particles (Binkowski and Shankar, 1995). For sulfate, nucleation from the gas phase is additionally considered (Kerminen and Wexler, 1994) and

particles are assigned to the soot-free Aitken mode (if). Hygroscopic growth of aerosols is based on ISORROPIA2 (Fountoukis and Nenes, 2007) for inorganic compounds and discussed in Athanasopoulou et al. (2013) for organic aerosol. As aerosol sinks, sedimentation and dry deposition (Riemer, 2002) and impaction scavenging (Rinke, 2008) by rain are considered. Nucleation scavenging is not considered.

The M7 aerosol scheme considers 4 hygroscopic and 4 hydrophilic log-normal modes, including a nucleation mode but

excluding giant modes. Tab. 1 compares the physical characteristics of these modes to the modes of MADE. M7 features a mode-reorganization routine that transfers the largest particles within a mode to the next larger mode if the modal radius exceeds the boundaries indicated in the table. M7 includes fewer chemical species than MADE. It transports DU, SS, BC, POA and $SO_4$ and thus especially omits nitrogen species and secondary organic aerosols. The chemical composition of M7 modes is illustrated and compared to MADE in Fig. 1. Inter-modal coagulation is considered for all modes, intra-modal coagulation

is neglected for the coarse modes (cs, ci) and accumulation mode dust (ai). Primary emissions are identical to MADE and in particular follow MADE size-distributions. They are assigned to M7 modes based on the mode correspondence shown in Tab. 1: BC is emitted into the insoluble carbon mode (ki), POA is partitioned in the same way as for MADE to the soluble




Aitken (ks) and accumulation mode (as). Giant dust and sea salt emission are ignored and accumulation and coarse mode dust emissions are assigned to the pure dust modes (ai, ci) in M7. Sulfate can nucleate into the nucleation mode (ns) (default scheme used in this study: Kazil and Lovejoy (2007), optional: Vehkamäki et al. (2002)) or condense onto the larger soluble modes (ks, as, cs). Hygroscopic growth of the soluble modes (nc, ks, as, cs) is based on $\kappa$-Köhler theory (Petters and Kreidenweis, 2007).

Aerosol removal by dry deposition and impaction scavenging follows the same parameterizations as for MADE.

## 2.2 Sulfur chemistry

As part of the full gas-phase chemistry, COSMO-ART considers the following sulfur oxidation reactions,

$$DMS + NO_3 \rightarrow SO_2 \tag{1}$$

$$DMS + HO \rightarrow SO_2 \tag{2}$$

$$DMS + HO \rightarrow 0.4\,DMSO + 0.6\,SO_2 \tag{3}$$

$$DMSO + HO \rightarrow 0.6\,SO_2 \tag{4}$$

$$SO_2 + HO \rightarrow SO_4 + HO_2 \tag{5}$$

where the reaction equations are restricted to prognostic species such that non-prognostic species have been omitted. Aqueous-phase chemistry, namely in-droplet oxidation of $SO_2(aq)$, is not included in the standard setup. DMS emissions are calculated

online based on wind speed (Nightingale et al., 2000). Anthropogenic gaseous emissions are based on inventory data. Dry deposition according to Baer and Nester (1992) and gas-to-particle conversion are considered as sinks of gas-phase species.

The efficient M7 chemistry consists of DMS, $SO_2(g)$, $SO_4(g)$ and $SO_4(aq)$ as interactive variables and requires external input for reactive oxidants HO, $O_3$, $NO_2$ and $H_2O_2$. To prescribe theses species, spatially heterogeneous monthly mean values are typically used. A steady-state value for $NO_3$ is additionally derived from the $NO_2$, $O_3$ and DMS input fields. The following

sulfur-oxidation reactions are considered

aqueous-phase chemistry: $$SO_2(aq) + H_2O_2(aq) \rightarrow SO_4(aq) \tag{6}$$

$$SO_2(aq) + O_3(aq) \rightarrow SO_4(aq) \tag{7}$$

day-time gas-phase chemistry: $$DMS(g) + HO(g) \rightarrow SO_2(g) \tag{8}$$

$$DMS(g) + HO(g) \rightarrow SO_4(g) \tag{9}$$

$$SO_2(g) + HO(g) \rightarrow SO_4(g) \tag{10}$$

night-time gas-phase chemistry: $$DMS(g) + NO_3(g) \rightarrow SO_2(g) \tag{11}$$

where non-prognostic products have been omitted. Day- and night-time reactions are exclusive and the seasonal variability of daylength is taken into account. Aqueous-phase chemistry requires the presence of cloud water but is independent of solar insolation. The dissolution of the gaseous species for the aqueous-phase reactions is based on the effective Henry constants

determined by the cloud droplets pH-value. Assuming that most cloud droplets have emerged from the activation of accumu-





lation mode aerosol, $SO_4(aq)$ resulting from the aqueous-phase reaction is in most cases assigned to the mixed accumulation mode (mode as in Figure 1 and Table 1).

### 2.2.1 Aerosol-radiation interactions

The optical properties of MADE and M7 aerosol particles, i.e. extinction coefficient, single-scattering albedo and asymmetry factor are parameterized based on Mie calculations. Optical properties of MADE aerosols are distinguished on a modal basis such that for each mode a representative refractive index is assumed and calculations are performed for modal diameters of emitted particles. The parameterization for mixed and anthropogenic modes is discussed by Vogel et al. (2009), for sea salt by Lundgren (2012) and for dust by Stanelle et al. (2010). In contrast to MADE, optical properties of M7 aerosol are species-based: The modal refractive index is the mass-weighted average of the refractive indices of the different species (Zubler et al., 2011). This method requires a look-up table of Mie properties, which also allows to consider the simulated modal diameters instead of the values at emission applied for MADE.

### 2.3 Aerosol-cloud interactions

The activation of aerosol particles to cloud droplets is described in Bangert et al. (2011, 2012). The CCN-spectrum is based on classical Köhler theory (Köhler, 1936) for hygroscopic aerosol (MADE modes if, ic, so, jf, jc, sa, sb, sc; M7 modes ns, ks, as, cs) and on adsorption theory (Kumar et al., 2011) for non-hygroscopic particles (MADE modes da, db, dc; M7 modes ki, ai, ci). Supersaturation follows from the parameterization of Nenes and Seinfeld (2003) and Fountoukis and Nenes (2005), which is based on adiabatic parcel ascent. For the updraft velocity, a probability density function (PDF) about the grid mean value is used. The parameterization takes into account the competition of different particles and solves the supersaturation balance equation based on population splitting into kinetically-limited and equilibrating activated aerosol particles. For cloud-base activation, entrainment of below-cloud aerosol is considered (Ghan et al., 1997). For in-cloud activation, the depletion of supersaturation by existing droplets is accounted for by treating these droplets as giant CCN following Barahona et al. (2010).

Ice nucleation is based on the empirical, surface-based INP-spectrum of Phillips et al. (2008), which does not distinguish different freezing modes. As an alternative, Ullrich et al. (2016) have recently derived and implemented nucleation spectra for immersion freezing of dust and deposition nucleation on dust and soot based on the ice-nucleation active site approach and measurements from the AIDA cloud chamber. Table 2 summarizes how INP-spectra are applied to MADE aerosols in the standard setup of COSMO-ART and to MADE and M7 aerosol for this study. The implementation of ice nucleation (Bangert et al., 2012) is based on Barahona and Nenes (2009a, b). For temperatures higher than the onset temperature of homogeneous freezing, i.e. $T > 235\,K$, grid-scale supersaturation with respect to ice is applied to determine the ice nucleation rate from the INP-spectrum. At lower temperatures, the competition of heterogeneous ice nucleation and homogeneous freezing of solution droplets is taken into account via the ice-supersaturation equation for an ascending parcel. For its updraft, a PDF about the grid mean value is applied.

The activation and ice nucleation parameterizations are coupled to a 2-moment microphysics scheme with 5 hydrometeor classes (cloud droplets, rain, ice crystals, snow and graupel) (Seifert and Beheng, 2006; Noppel et al., 2010). This scheme





does not distinguish between warm, mixed-phase and cirrus clouds but its processes are based on temperature, saturation, and liquid and ice water content in the respective grid box. We will therefore use the term *liquid cloud* or *warm cloud* to denote cloudy regions without cloud ice, *mixed-phase cloud* to denote cloudy regions in which both, cloud liquid and cloud ice, are present, and *ice-phase cloud* for regions which contain cloud water in the form of ice but no liquid. The latter may correspond to glaciated clouds or to cirrus clouds. We reserve the expression *cirrus* for ice-phase clouds at temperatures lower than 235 K, in which homogeneous freezing of solution droplets occurs.

The coupling of the activation- and ice-nucleation routines to the cloud microphysics scheme is adapted from the standard setup of COSMO-ART and identical for both aerosol schemes in this study. As for the standard version of COSMO-ART, neither liquid nor ice-phase nucleation scavenging is considered. The coupling of the parameterized number of activated aerosol particles to microphysics in the standard setup of COSMO-ART is based on the assumption that in-cloud activation is largely inhibited by the depletion of supersaturation on pre-existing cloud droplets. CCN-depletion is only accounted for by limiting the number of cloud droplets to the total number of soluble Aitken and accumulation mode particles. In this study, CCN-depletion is taken into account by subtracting the number of existing cloud droplets from the number of newly activated droplets predicted by the activation parameterization.

In the standard setup, ice nucleation in mixed-phase as well as ice-phase clouds is coupled to the cloud microphysics scheme based on the assumption that ice nucleation converts water vapor into ice. Ice nucleation in mixed-phase clouds is thus assumed to proceed purely by condensation nucleation (Table 2). For mixed-phase clouds in this study, we assume that immersion and contact freezing convert cloud droplets into ice crystals such that droplet number concentration and mixing ratio are reduced by mixed-phase ice nucleation. Ice-nucleation in ice-phase clouds follows the previous approach of MADE and depletes water vapor. Unmodified from the standard setup, INP depletion is accounted for by a number adjustment that subtracts the existing number of ice crystals and snow flakes from the crystal number predicted by the parameterization.

## 3 Setup

Simulations for this study are performed for a Saharan dust outbreak reaching Europe in May 2008. The domain covers the dust sources in Northern Africa and extends to Western and Central Europe (Figure 2). The model setup has a horizontal resolution of 25 km at a time step of 30 s. The vertical resolution decreases with heigh, starting with 20 m in the surface layer and reaching 1000 m at the model top corresponding to a height of 22 km. We simulate a 90 h period, starting on May 22nd, 00:00. To allow for spin-up of aerosol concentrations, we analyze the time-average of hourly output from the last 24 h of the simulation.

Meteorological initial and boundary conditions are provided by the global model GME (Majewski et al., 2002). For the full ART chemistry, initial and boundary conditions of gases with the exception of DMS, $SO_2$, and $SO_4$ are based on the global chemistry model MOZART (Emmons et al., 2010). For DMS, $SO_2$, $SO_4$ and aerosols, no initial and boundary conditions are provided. Anthropogenic emissions follow the TNO/MACC inventory (van der Gon et al., 2010; Kuenen et al., 2011), which does not provide emissions for Africa. Surface properties for parameterized emissions rely on the GLC2000 dataset (Bartholomé and Belward, 2005) and on Marticorena et al. (1997) for dust.





**Table 2. Coupling of aerosol modes to ice nucleation parameterizations.** The table summarizes which ice nucleation modes are considered for the pure dust and soot modes and modes with dust and/or soot core, depending on the aerosol scheme and ice nucleation parameterization. In the standard setup of COSMO-ART, the condensation freezing parameterization, which takes into account MADE aerosol, is combined with a droplet freezing routine from the cloud microphysics scheme, which is not coupled to MADE. Homogeneous freezing of solution droplets follows Barahona and Nenes (2009b).

| | pure DU | coated DU | pure BC | coated BC | dissolved aerosol | |
| --- | --- | --- | --- | --- | --- | --- |
| | | | | | without core | with core |
| MADE modes | da, db, dc[1] | | so | ic, jc | if, jf, sa, sb, sc | ic, jc |
| M7 modes | ai, ci | as, cs | ki | ks, as, cs | (ns)[2] | ks, as, cs |

**COSMO-ART with Phillips et al. (2008) (standard)**

| | | | | | |
| --- | --- | --- | --- | --- | --- |
| ice-phase/cirrus | deposition | | | homogeneous | - |
| mixed-phase | immersion (Bigg (1953), not coupled to MADE) condensation | | | - | |

**MADE and M7 with Phillips et al. (2008) (this study)**

| | | | |
| --- | --- | --- | --- |
| ice-phase/cirrus | deposition | | homogeneous |
| mixed-phase | immersion + contact | | - |

**MADE with Ullrich et al. (2016) (this study)**

| | | | |
| --- | --- | --- | --- |
| ice-phase/cirrus | deposition | - | homogeneous |
| mixed-phase | immersion | - | - |

**M7 with Ullrich et al. (2016) (this study)**

| | | | | |
| --- | --- | --- | --- | --- |
| ice-phase/cirrus | deposition | - | deposition | - | homogeneous |
| mixed-phase | - | immersion | - | | - |

[1] MADE features only uncoated dust that is interpreted as having a coating in the context of immersion freezing.

[2] The nucleation mode is the only soluble mode without core in M7. It is considered too small for homogeneous freezing.

Table 3 summarizes the 6 different model settings used for this study. Simulations **sim**, **simSIG**, **passive** and **coupled** are performed with both, MADE and M7. Simulations **simAQ** and **simCL** are specific to and only performed for M7 such that overall 10 simulations have been performed.

Aerosol-radiation interactions are disabled for all simulations, aerosol-cloud interactions are restricted to simulations **cou-5 pled**. All other simulations thus feature passive aerosols such that the simulated meteorology is identical for simulations with MADE and M7. Without aerosol-cloud interactions, the 2-moment cloud microphysics is not required. We therefore employ the operational 1-moment scheme (Reinhardt and Seifert, 2006) in simulations with passive aerosol.





Simulations **sim** aim to make the model setup of M7 and MADE as similar as possible: The M7-only aqueous-phase chem-istry, the MADE-only giant modes and SOA, $NO_3$, $NH_4$ and unspeciated PM2.5 as MADE-only species are disabled; a uni-versal standard deviation of $\sigma_{universal} = 1.7$ is used for all MADE and M7 modes instead of the default standard deviations indicated in Table 1; for the oxidant fields required by the M7 chemistry hourly outputs of the respective fields from MADE

simulations are used instead of climatological values. Simulations **sim** aim to investigate the sensitivities of aerosol burden, aerosol size distribution and gas-phase chemistry without taking into account the disabled structural differences.

Simulations **passive** correspond to default setups of MADE and M7 and allow to explore additional sensitivities arising from aqueous-phase chemistry, climatological oxidant fields, different modal standard deviations and additional aerosol species. For these simulations, we additionally investigate the optical and cloud- and ice-forming properties of the aerosol distributions by

offline diagnostics: Routines for optical properties, droplet activation and ice nucleation are called without passing the results on to the cloud microphysics and radiation scheme of the model. The ice nucleation routine is called in mixed-phase setting when the 1-moment cloud microphysics scheme predicts both, cloud ice and cloud water, and in ice-phase setting when cloud water is absent. The activation routine is applied in its setting for new cloud formation, i.e. without cloud-base entrainment of aerosol and without considering supersaturation depletion by existing droplets. It is called in all grid boxes where cloud

water is predicted by the 1-moment scheme. For computational reasons, the updraft PDF is replaced by applying an updraft $w^* = w + 0.8\sqrt{TKE}$ where $w$ is the grid-scale updraft and TKE denotes the subgridscale turbulent kinetic energy (Bangert, 2012).

Simulations **simSIG**, **simAQ** and **simCL** feature settings intermediate to **sim** and **passive** and are intended to individually investigate the effects of modal standard deviation, aqueous-phase chemistry or climatological oxidant fields, respectively.

Simulations **coupled** with 2-moment microphysics and aerosol-cloud coupling are conducted to investigate the relationship between CCN, INP, cloud droplet and ice crystal numbers.

## 4   Results from the sensitivity experiments

Figure 2 illustrates the dominant transport patterns for aerosols on the analysis day: Following the transport from Africa over the Mediterranean to central Europe, the flow turns to a low-pressure system off the Bay of Biscay. The corresponding M7

aerosol burdens of sea salt, dust, BC and POA are illustrated in Figure 3 (left column): Dust is transported from the Saharan source regions over the Mediterranean Sea to the southern parts of Germany and France. Sea-salt containing maritime air is advected over most of the domain, with the exception of Eastern Africa. Strong winds south of Britain explain strongest sea salt emissions and burdens in this region. For the African part of the domain, no anthropogenic emissions are available. Accordingly, BC and POA are largely restricted to continental Europe, the Mediterranean Sea and the Atlantic part of the

domain. The corresponding $SO_4$ burden is depicted in Figure 4 (middle row). It is restricted to the northern and western half of the domain because continental Africa does neither provide anthropogenic emissions of $SO_2$ nor natural DMS-derived sulfate. In some parts of the following analysis we distinguish different regions based on aerosol composition (Figure 2): The region denoted as 'Atlantic' comprises maritime regions at which surface dust is absent, the expression 'Mediterranean', in





**Table 3. List of simulations.** A 'y' shows that a model feature is active, 'n' indicates it is not active. See main text for details.

| simulation setting | applicable aerosol schemes | aerosol-cloud coupling | diagnostic CCN and INP | default standard deviation | M7 aqueous-phase chemistry | M7 climatological oxidant fields | MADE giant modes & additional species | subject of investigation |
|---|---|---|---|---|---|---|---|---|
| **sim** | M7 & MADE | n | n | n | n | n | n | gas-phase chemistry, aerosol burden, size distributions |
| **simSIG** | M7 & MADE | n | n | y | n | n | n | modal standard deviation |
| **simAQ** | M7 | n | n | n | y | n | n | aqueous-phase chemistry |
| **simCL** | M7 | n | n | n | y | y | n | oxidant fields |
| **passive** | M7 & MADE | n | y | y | y | y | y | comparison of default setups, radiative and cloud-forming properties |
| **coupled** | M7 & MADE | y | y | y | y | y | y | effect of CCN/INP differences on cloud droplet and ice crystal number |

contrast, characterizes dusty maritime regions. 'Europe' stands for continental ares with anthropogenic emissions and 'Africa' for continental sites without anthropogenic emission.

Aerosol burdens for MADE and M7 agree within 10% (Figure 3, Table 4), which confirms our strategy for simulations **sim** in choosing the setup such that MADE and M7 are very similar. Dust burdens are identical for M7 and MADE: The transfer

of dust into the soluble M7 modes via condensation is ineffective (coagulation is neglected due to large particle sizes, Sec. 2) such that MADE and M7 both describe dust by two identical pure modes. The low coating in simulation **sim** is a result of a general underestimation of sulfate for the this setup (cf. Table 6). The M7 sea salt burden is increased by ca. 10% as compared to MADE while the SO$_4$ burden is by ca. 10% decreased. BC and POA burden are decreased by less than 5%. The following discussion of simulations **sim**, **simSIG**, **simAQ** and **simCL** is greatly facilitated by this similarity.

## 4.1    Sensitivities of aerosol size distributions and removal

Primary emissions are identical for simulations with MADE and M7 (Sect. 2) such that differences in primary aerosol burdens are attributable to the aerosol sinks, i.e. dry deposition and impaction scavenging. Differences in sulfate burden between MADE and M7 are likewise dominated by differences in removal and not in the sulfate production rate (Figure 4).

The efficiency of both removal processes depends on particle size and becomes inefficient if particle radii approach the

Greenfield gap at 0.1 μm. Whether a shift of the size-distribution results in increased or decreased removal depends on its





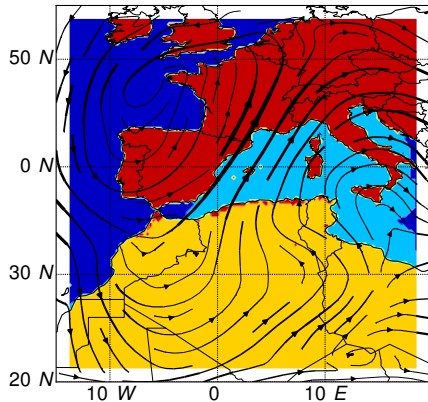

**Figure 2. Aerosol mass transport** as represented by the weighted vertical average $\langle x \rangle = \sum_i w_i x_i / \sum_i w_i$ of the horizontal wind field $x$ where weights $w$ are given by the total dry aerosol mass concentration. Wind direction is indicated by arrow heads and its strength encoded in line thickness where the thickest lines correspond to $40\,\mathrm{ms}^{-1}$. The background colors illustrate the geographic regions Africa (yellow), Mediterranean Sea (light blue), Europe (red) and Atlantic (dark blue). See main text for details of region definitions.

relative position to the Greenfield gap: A shift of an Aitken mode to smaller sizes or of a coarse mode to larger sizes enhances removal. The effect of an accumulation mode shift depends on the details of the Greenfield gap and cannot easily be predicted. Removal is dominated by impaction scavenging in the cloudy northern half of the domain, where the abundance of sulfate, sea salt, BC and POA is largest. Only for dust, dry deposition is important, especially in the African source regions (not shown).

### 4.1.1 Sensitivity of size distribution to modal composition

Figure 5 depicts domain-averaged volume size-distributions of different species for MADE and M7 obtained from simulations **sim** (red). The size-distribution of M7 sea salt is shifted to smaller particle sizes as compared to MADE. This is a result of the internal mixture of sea salt in M7 as compared to the externally mixed sea salt modes of MADE (Table 1): Sea salt emissions only contribute a fraction of the total number of particles in the M7 mixed modes such that the average sea salt mass per mixed aerosol particle is reduced as compared to the average mass of emitted particles and the corresponding size of MADE sea salt. For dust, not only the burdens but also the size distributions are effectively identical for MADE and M7.

While MADE-sulfate is found in a single broad peak of a large Aitken or small accumulation mode, M7-sulfate mass shows a distinct trimodal structure. The position of the pronounced M7 sulfate coarse mode corresponds to that of coarse-mode sea salt. As dust is hardly coated and BC and POA are not abundant in the coarse-mode size range, the mixed M7 coarse mode corresponds to sulfate-coated sea salt. In contrast, the MADE sea salt coarse mode is not significantly coated. The M7 coarse-mode coating could be more effective because particles are smaller and provide a larger surface. In addition, while MADE sulfate is restricted to condensation as process for transfer into the coarse mode, M7 sulfate can additionally be transferred





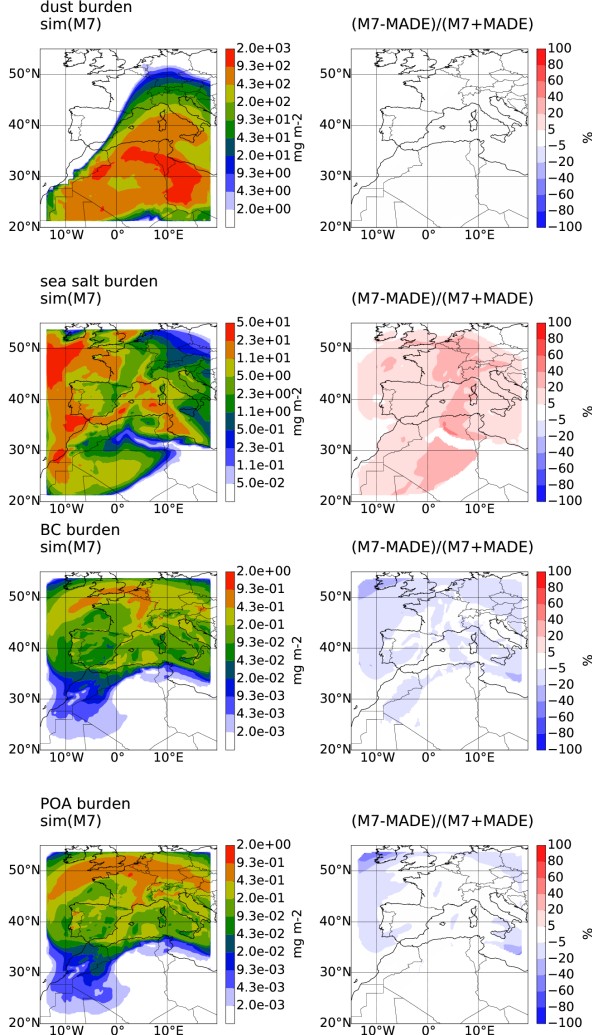

**Figure 3. Aerosol burdens** of dust (first row), sea salt (second row), BC (third row) and POA (last row) for M7 (left, data points exceeding the scale have been clipped to the maximum value) and differences to MADE (right, to prevent diverging values, percentage differences $(f_1 - f_2)/(f_1 + f_2)$, $f_1$: M7, $f_2$: MADE, are only determined for data points with $f_{1/2}(\text{lat}, \text{lon}) > 0.01 \cdot P_{95}(f_1)$, where lat and lon denote latitude and longitude of the horizontal position and $P_{95}$ the 95th-percentile of all data points in the domain) for simulations **sim**. Unless explicitly mention otherwise, aerosol burden refer to dry aerosol mass.

from the accumulation to the coarse mode by mode reorganization once the median radius exceeds the maximum value for its mode.

The separated Aitken and accumulation mode peaks in M7 sulfate as compared to the single peak for MADE correspond to the BC and POA size distributions: M7 BC is located at smaller, M7 POA at larger sizes than for MADE. The location

5   of the M7 POA peak corresponds to the M7 accumulation-mode sea-salt peak and indicates that POA-containing particles in





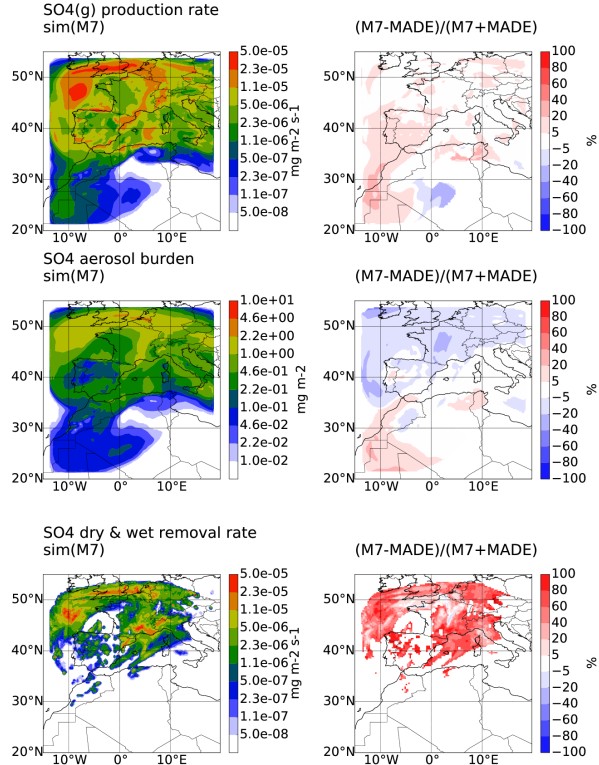

**Figure 4. Sulfate budget for simulations sim.** Comparison of M7 (left column) to MADE (percentage-difference plots in the right column) in terms of the vertical integral of the gas-phase production rate of $SO_4$ (first row), sulfate burden (second row) and the sum of dry deposition and vertically integrated impaction scavenging rate of sulfate (last row). See Figure 3 for plot details.

M7 are enlarged by internal mixture with sea salt. The increase in MADE accumulation mode BC as compared to M7 likely results from different strategies to describe growth by condensation in MADE and M7: For both schemes, BC is emitted into a pure Aitken mode and rapidly coated. In M7, coated BC is assigned to the mixed Aitken mode and can subsequently be transferred to the accumulation mode by mode reorganization (Vignati et al., 2004). The extent of the mode reorganization is

5   not directly coupled to the size of the coated soot particles but to the characteristics of the mixed Aitken mode with sizes e. g. being influenced by the transfer of small particles from the nucleation mode or transfer of large particles to the accumulation mode. The MADE coating routine directly assigns a fraction of the newly coated BC to the accumulation mode (Riemer, 2002).

The differences in aerosol burdens between MADE and M7 can be traced back to the size distributions: The M7 sulfate burden is decreased in comparison to MADE in the northern part of the domain due to increased removal of M7 coarse-mode

10   sulfate. The burden of M7 sea salt is increased due to the smaller size of the mixed coarse mode as compared to the MADE sea salt coarse mode, which results in less efficient impaction scavenging for M7. The BC burden of M7 is smaller than that of MADE because of increased removal due to smaller sizes of BC-containing particles in M7. The decrease in M7 POA burden





**Table 4. Horizontal averages of relative differences** $\Delta = (M7 - MADE)/(M7 + MADE)$ between MADE and M7 in percent for simulations **sim** and **passive**. Values correspond to Figures 4, 3, 8, 9 and 10, where production rates are vertically integrated and concentrations are vertically averaged. With the exception of SIA and accumulation and coarse mode burden, aerosol burden and $SO_4$ production and removal are not illustrated for simulation **passive**. AOD and CCN are not shown for simulation **sim**. If not explicitly stated otherwise aerosol burden correspond to dry aerosol mass.

|  | sim | passive |
|---|---|---|
| $SO_4$ production rate (domain average) | 4 | 98 |
| $SO_4$ aerosol burden (domain average) | - 4 | 67 |
| $SO_4$ aerosol burden (average over upper left quadrant of domain) | -11 | 68 |
| $SO_4$ dry & wet removal rate (domain average) | 61 | 76 |
| SIA burden (average over Atlantic) | -11 | -20 |
| SIA burden (average over Mediterranean Sea) | 0 | 15 |
| dust burden (domain average) | 0 | -44 |
| sea salt burden (domain average) | 12 | 27 |
| OA burden (domain average) | - 3 | -18 |
| BC burden (domain average) | - 6 | - 4 |
| AOD (average over upper left quadrant of domain) |  | -54 |
| AOD (average over lower right quadrant of domain) | 9 | -10 |
| wet accumulation and coarse mode burden (average over upper left quadrant of domain) |  | -28 |
| wet accumulation and coarse mode burden (average over lower right quadrant of domain) | 0 | -11 |
| CCN number concentration in liquid clouds (domain average) |  | -54 |
| CCN number concentration in mixed-phase clouds (domain average) |  | -57 |

can be explained by the position of the M7 mixed accumulation mode being shifted away from the Greenfield gap as compared to the MADE accumulation modes.

### 4.1.2 Sensitivity of size distribution to modal standard deviation

The effect of the modal standard deviation $\sigma$ on the size distribution is illustrated in Figure 5. Plotting the data of simulations
5    **sim** with the default standard deviations of the aerosol schemes (green, see Table 1 for values of $\sigma_{\text{default}}$) instead of the universal standard deviation $\sigma_{\text{universal}} = 1.7$ used to generate the data illustrates the structural effect of the standard deviation as opposed to the effects arising from the influence of $\sigma$ on aerosol microphysical processes. The structural effect is most pronounced for dust: With $\sigma_{\text{da}} = 1.7 = \sigma_{\text{universal}}$, the width of the MADE accumulation mode remains unchanged, while the MADE coarse mode becomes slightly narrower with $\sigma_{\text{db}} = 1.6$. For M7, the default accumulation mode is narrowed ($\sigma_{\text{ai}} = 1.59$) and the
10    default coarse mode broadened ($\sigma_{\text{ci}} = 2$).



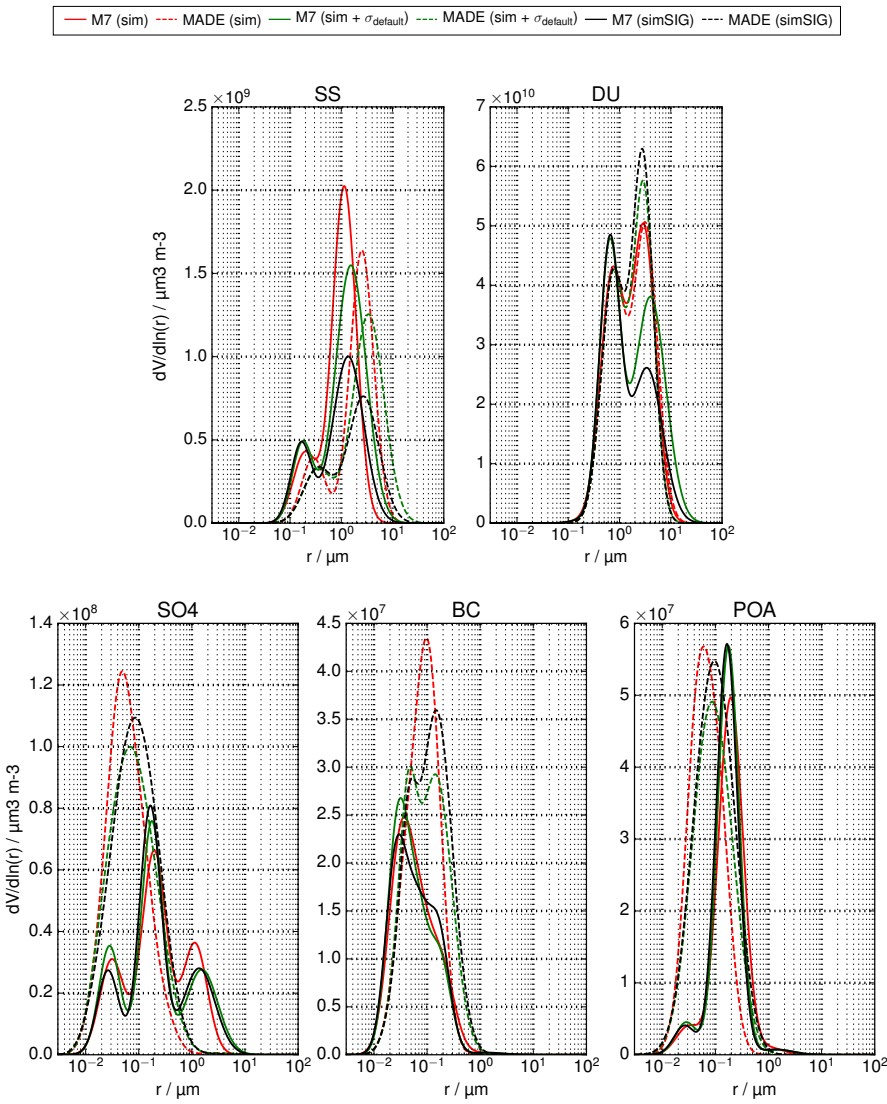

**Figure 5. Domain-averaged volume distributions for different species and standard deviations.** Distributions are obtained by weighting the total dry volume distribution by the fraction the respective species contributes to the total mass. The total distribution is the sum of all model modes, which are determined from the vertical sum and horizontal averages of the corresponding dry masses and numbers. Species include sea salt (SS), dust (DU), sulfate (SO$_4$), soot (BC) and primary organic carbon (POA). The figure compares simulations **sim**, generated with the universal standard deviation $\sigma_{\text{universal}} = 1.7$ for all modes (red), simulations **sim** but plotted using default standard deviations $\sigma_{\text{default}}$ as given in Table 1 (green) and simulations **simSIG**, generated with $\sigma_{\text{default}}$ (black).

The effect of $\sigma$ on aerosol microphysics can be assessed by comparing the differences between simulation **sim** plotted with default standard deviations (green) and simulations **simSIG** where the size distributions were generated using the default





standard deviations (black). Effects are strongest for the coarse modes of dust and sea salt. Dust mass in the coarse mode is determined by the efficiency of dry deposition, which is the dominant removal process in the cloud-free African source regions. The sedimentation velocity of a log-normal mode is given by $v_{\mathrm{sedi}} \propto r^2 \exp(8\ln^2 \sigma)$ (Slinn and Slinn, 1980) such that dry deposition is increased for larger $\sigma$ which corresponds to an increased number of very large particles. The dust burden of

MADE accordingly increases by 2% when applying the smaller default standard deviation while the M7 dust burden decreases about 10% for the enlarged $\sigma_{\mathrm{ci}}$. A similar argument for the impaction scavenging of coarse mode sea salt explains a 20% decreases in MADE and M7 sea salt burden when using $\sigma_{\mathrm{default}} = \sigma_{\mathrm{cs}} = \sigma_{\mathrm{sb}} = 2$ instead of $\sigma_{\mathrm{universal}}$.

## 4.2   Sensitivity of chemical sulfate production to aqueous-phase reactions and oxidant fields

Figure 4 compares the chemical sulfate production as sources of atmospheric sulfate arising from MADE and M7 aerosol

dynamics with full and efficient gas-phase sulfate chemistry for simulations **sim** (recall from Sections 2.2 and 3 that nitrate chemistry is not considered). With domain-averaged differences below 5% (Table 4), the M7 gas-phase chemistry (Equations 7 - 11) and the ART-chemistry (Equations 1 - 5) are equally efficient in producing $SO_4$.

The importance of aqueous-phase chemistry as a source of atmospheric sulfate aerosol is illustrated in Figure 6. The aqueous-phase reaction rate in simulation **simAQ** is about 2 times larger than the gas-phase reaction rate in simulation **sim** without

aqueous-phase chemistry (compare Figures 4 and 6). As the occurrence of $SO_2$ coincides with cloudy conditions in the northern and western part of the domain, the aqueous-phase reaction efficiently consumes $SO_2$ and leads to a 40% reduction of its concentration as compared to simulation **sim** (Table 5). The gas-phase reaction rate in simulation **simAQ** is reduced by 50% in comparison to **sim** due to the competition with the aqueous-phase reaction for $SO_2$. The resulting sulfate burden of **simAQ** is 70% increased as compared to **sim**.

The use of monthly-mean climatological oxidant fields instead of hourly values simulated by the full ART gas-phase chemistry influences the sulfate burden by less than 5% (Figure 6, Table 5). The almost identical results in our case are the consequence of compensating effects on the aqueous-phase reaction rates, which dominates total sulfate production: A 20% reduction in the $H_2O_2$ climatological oxidant field as compared to the detailed chemistry is largely compensated by a 6% increase in $O_3$ (Table 5) and thus only results in a 1% reduction of aqueous-phase production of sulfate (Figure 6, Table 5). The gas-phase

production rate of sulfate exhibits an inconsequential signal, which probably emerges from the interplay of enhancing effects of a locally dampened aqueous-phase reaction rate and dampening effects of decreases in the climatological concentrations of OH and $NO_2$ by 30 and 40%, respectively, as compared to the hourly values (Figure 6, Table 5).

The effect of the different chemistry setups on the sulfate level is summarized in Table 6, which compares the average surface concentrations of $SO_4$ over continental Europe. According to e. g. Fountoukis et al. (2011), concentrations about $1 - 2\,\mu\mathrm{g\,m}^{-3}$

are expected. These values are not reached with gas-phase sulfate chemistry alone, but require the efficient aqueous-phase reaction, which is consistent with the findings of previous studies and especially by Knote et al. (2011).





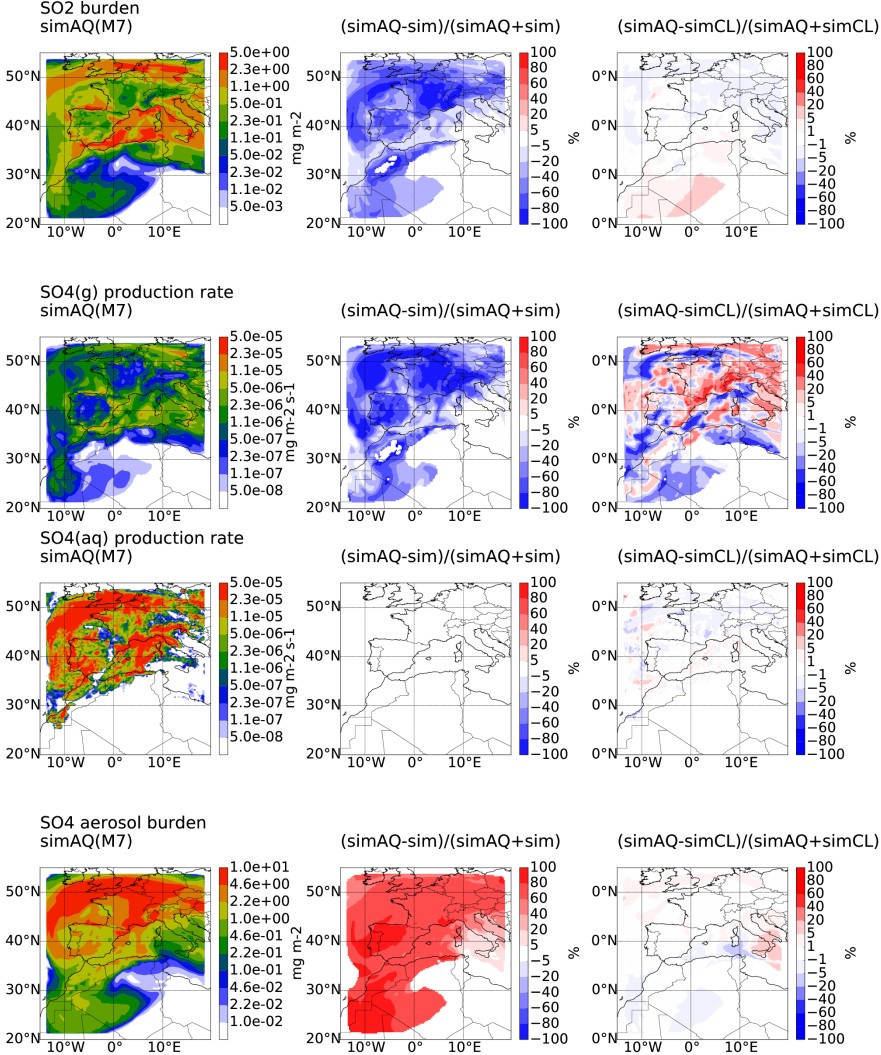

**Figure 6. Sulfate production from aqueous-phase chemistry and using climatological oxidant fields.** The figure compares simulation **simAQ** (left) to simulations **sim** (percentage-difference plots in the middle) and **simCL** (percentage-difference plots on the right, note the different colorscales). The first and fourth row shows the burden of $SO_2$ and sulfate aerosol. The second and third rows depict vertical integrals of gas- and aqueous-phase production rates of $SO_4$. See Figure 3 for plot details.

## 5   Results from comparison of default setups

Differences in the sulfate budgets of MADE and M7 in their default configuration (**passive** simulations according to Table 3) are dominated by the M7-only aqueous-phase chemistry (Table 4). As discussed in the previous section, aqueous-phase chemistry is about twice as efficient in oxidizing $SO_2$ as the gas-phase chemistry. The M7 sulfate burden is about 70% increased for





**Table 5. Horizontal averages of relative differences** $\Delta = (\text{simAQ} - \mathbf{x})/(\text{simAQ} + \mathbf{x})$ between different M7 chemistry setups in percent for simulation **simAQ** in comparison to simulations $\mathbf{x} = \text{sim}, \text{simCL}$. Values of $SO_2$ and $SO_4$ correspond to Figure 6, where the productions rates are vertically integrated. Differences in oxidant fields are based on weighted vertical averages as in Figure 2, with the gas-phase production rate of $SO_4$ as weight for the gas-phase oxidant OH and gas-phase oxidant precursor $NO_2$ and weighted with the aqueous-phase reaction rate for the aqueous-phase oxidants $H_2O_2$ and $O_3$. Where the sign of a difference signal is not uniform throughout the domain, representative quadrants have been chosen. Oxidant fields are not illustrated.

|  | sim | simCL |
|---|---|---|
| $SO_2$ burden (domain average) | -39 | 0 |
| $SO_4(g)$ production rate (domain average) | -47 | - 3 |
| $SO_4(aq)$ production rate (domain average) | 100 | - 1 |
| $SO_4$ aerosol burden (domain average) | 70 | 0 |
| OH (average over upper right quadrant of domain) | 0 | 31 |
| OH (average over lower left quadrant of domain) | 0 | -10 |
| $NO_2$ (domain average) | 0 | 42 |
| $O_3$ (domain average) | 0 | - 6 |
| $H_2O_2$ (average over upper right quadrant of domain) | 0 | 20 |

**Table 6. $SO_4$ surface concentrations** in $\mu\text{gm}^{-3}$ for simulations according to Tab. 3. Values are horizontal averages over continental Europe.

|  | sim | simAQ | simCL |
|---|---|---|---|
| M7 | 0.23 | 1.66 | 1.65 |
| MADE | 0.27 | - | - |

aqueous- and gas-phase chemistry (**simAQ**) as compared to gas-phase chemistry alone (**sim**). In simulation **sim**, the different gas-phase chemistries for M7 and MADE result in almost identical sulfate burdens. Consequently, when comparing M7 with gas- and aqueous-phase chemistry to MADE in **passive** simulations, a 70% increase for M7 is observed (Table 4). Comparing the size distribution of M7 sulfate mainly produced by aqueous-phase chemistry (Figure 7) to that produced by the gas-phase

5  reaction (Figure 5) illustrates that the aqueous-phase chemistry deposits sulfate mainly into the accumulation and to a lesser extent into the coarse mode, while gas-phase chemistry additionally transfers sulfate to Aitken modes particles via condensation or coagulation with nucleation-mode particles. Note that to consider the additional MADE aerosol species in the comparison with M7, we combine sulfate, nitrate and ammonium into a secondary inorganic aerosol (SIA) class. For M7, SIA is identical to sulfate aerosol.

10  The higher M7 sulfate burden is compensated for by MADE nitrate and ammonium when studying SIA (Figure 8). In the Atlantic part of the domain, overcompensation occurs and the SIA burden is reduced by about 20% for M7 as compared to



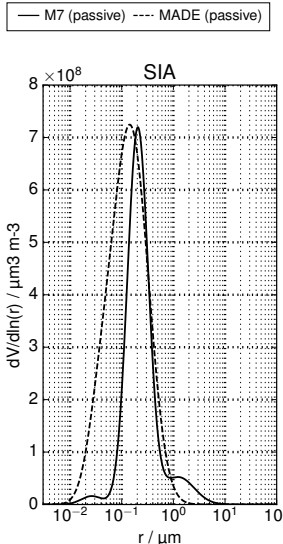

**Figure 7. Domain-averaged volume distributions of SIA for default setups.** This figure corresponds to $SO_4$ in Figure 5 but shows SIA for the **passive** simulations. SIA (secondary inorganic aerosol) corresponds to $SO_4$ for M7 and additionally includes $NO_3$ and $NH_4$ for MADE.

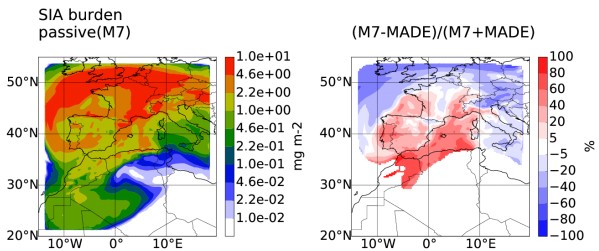

**Figure 8. SIA burden for default setups.** This figure corresponds to $SO_4$ in Figure 3 but shows SIA for the **passive** simulations. SIA (secondary inorganic aerosol) corresponds to $SO_4$ for M7 and additionally includes $NO_3$ and $NH_4$ for MADE.

MADE (Table 4). The SIA burden in the central (Mediterranean) part of the domain remains about 15% increased for M7 in comparison to MADE.

The sea-salt size distributions of MADE and M7 from **passive** simulations (not shown) are qualitatively similar to simulation **sim** (Figure 5). The impaction scavenging efficiency of sea salt remains higher for MADE than for M7. This effect is not compensated for by additional sea salt mass in the MADE giant mode, keeping the sea salt burden of M7 enhanced as compared to MADE (Table 4). The importance of MADE giant sea salt is probably limited because the main emission regions of sea salt coincide with rainy regions such that most particles are immediately removed by impaction scavenging.

In contrast to simulations **sim**, the **passive** M7 dust burden is decreased by about 40% in comparison to MADE due to increased dry deposition of the wider coarse mode and because the M7 dust burden has no contribution from the giant mode.





The additional MADE dust leads to strongly enhanced difference between MADE and M7 in the height of the coarse/giant mode peak in the size distribution (not shown) that otherwise remains qualitatively similar to that from simulation **simSIG** (Figure 5, black).

Differences in BC burden remain similar to simulation **sim** (Table 4) as does the BC size distribution (not shown, but see
Figure 5). Similar to SIA, SOA and unspeciated aerosol from MADE is considered as part of an organic aerosol (OA) class. For M7, OA is identical to POA. SOA and unspeciated aerosols enhance the MADE OA burden to a 20% increased value as compared to M7 (Table 4). The OA size distribution (not shown) is qualitatively similar to that of POA from simulation **sim** (Figure 5).

### 5.1   Radiative properties

Figure 9 compares 550 nm-AOD for the **passive** simulations of MADE and M7. Comparing the pattern of AOD to the species burden in Figure 3 shows that it is dominated by dust over Africa and the Mediterranean region. Over continental Europe and the Atlantic part of the domain, AOD is controlled by anthropogenic aerosols and sea salt. MADE AOD is 10% enlarged as compared to M7 in the dust-dominated part of the domain and about 50% enhanced in the rest of the domain (Table 4). In the regions of strongest flow (Figure 2) differences of up to 100% occur (Figure 9). The increased MADE AOD can be attributed
to the additional modes and species of MADE, i.e. the giant dust mode, nitrate, ammonium, SOA and unspeciated aerosol: The AOD difference pattern is matched by the difference pattern of the total wet aerosol burden in the accumulation and coarse mode size ranges (Figure 9), which dominate the radiative effect because particles sizes correspond to the considered wavelength of 550 nm. The difference pattern between MADE and M7 accumulation and coarse mode soot does not correspond to AOD differences pattern (not shown) and confirms that differences are caused by the additional scattering MADE species
and not by the differences in the distributions of absorbing soot in the Aitken and accumulation modes in MADE and M7 (Figure 5).

Differences in radiative properties between MADE and M7 are dominated by differences in burden arising from the structural differences and not by differences in the parameterization of optical properties (Section 2.2.1). An estimate of the latter can be obtained from simulation **sim**: For this setup, dust burden and size distribution are identical for MADE and M7, and dust is the
only aerosol species over Africa in the lower left quadrant of the domain (Figures 3, 5). A 9% increased M7 AOD as compared to MADE in this region is thus caused by differences in the parameterization of aerosol optical properties alone. The decreased M7 AOD for simulation **passive** (Figure 9) illustrate that this parameterization effect is less important than the structural effect of the additional MADE giant dust mode.

### 5.2   Droplet-activation properties

MADE produces 50% more CCN than M7 (Figure 10, Table 4). The number distribution of soluble aerosols depicted in Figure 11 (a) illustrates that the increase in MADE CCN corresponds to an increased number of MADE aerosol particles in the Aitken mode size range that are large enough for activation as measured by a threshold radius of 35 nm based on the empirical activation parameterization by Lin and Leaitch (1997): MADE on the one hand features more particles in the Aitken size range




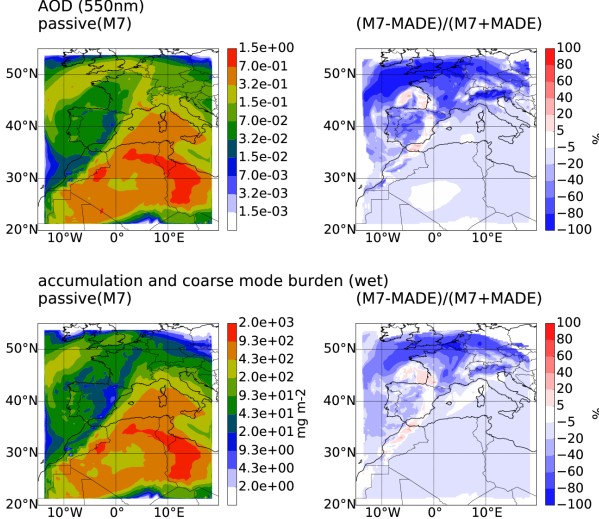

**Figure 9. Aerosol optical properties.** Aerosol optical depth at a wavelength of 550 nm (top) and total wet aerosol burden in accumulation and coarse modes (bottom) resulting from M7 (left) and MADE (percentage-difference plots on the right) aerosol for simulation **passive**. See Figure 3 for plot details.

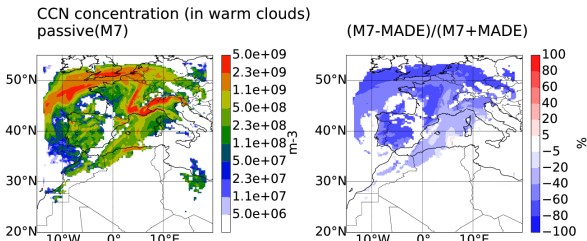

**Figure 10. Liquid-cloud CCN** concentrations derived from aerosol compositions predicted by M7 (left) and MADE (percentage-difference plot on the right) for **passive** simulations. Contours show vertical averages of concentrations in grid boxes that contain liquid-phase but no ice-phase cloud water.

due to additional emissions of unspeciated PM2.5 particles that are not considered in M7. On the other hand, MADE Aitken mode particles are larger due to additional coating from SOA, nitrate and ammonium. Note that aqueous-phase formed M7 sulfate cannot compensate for these species because it is predominantly assigned to accumulation mode particles, which are already large enough to be activated.

5    Figure 10 illustrates the situation for liquid clouds. Relative changes are comparable in mixed-phase clouds (Table 4), while absolute CCN numbers are about 20% lower than in liquid clouds (not shown) due to a general decrease in aerosol number concentration with height. Also note the CCN predicted in the absence of soluble Aitken and accumulation mode particles in the lower right quadrant of the domain (Figure 3): These result from adsorption activation of hydrophilic dust.





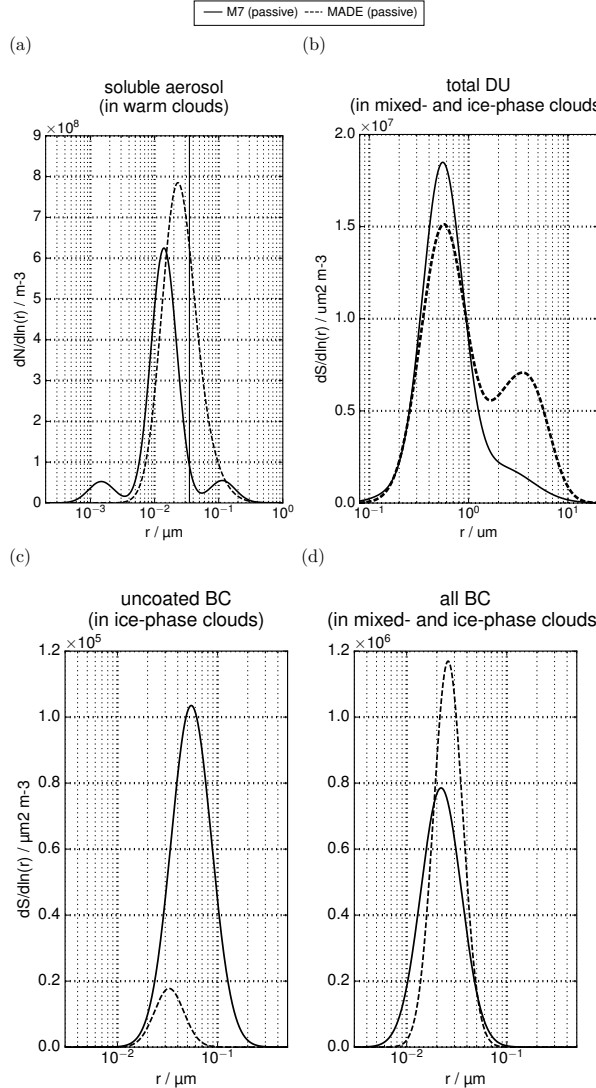

**Figure 11. Size distributions of cloud-active aerosol** for simulation **passive** using MADE and M7. Shown are the CCN-relevant number distribution of soluble aerosol in liquid clouds (a) as well as surface distributions of dust (b) and uncoated (c) and total soot (d) in the cloud regimes where these species serve as INP (Table 2).

## 5.3   Ice-nucleation properties

Dust and soot are considered as ice-nucleation-active species in our simulations (Section 2.3). Dust has a much higher ice-nucleation potential so that it dominates ice nucleation when present. This is the case over Africa and the Mediterranean Sea (Figure 3). Soot determines ice nucleation in the Atlantic part of the domain that is not affected by the dust outbreak.



**Table 7.** Horizontal averages of relative differences $\Delta = (\mathrm{M7}-\mathrm{MADE})/(\mathrm{M7}+\mathrm{MADE})$ between MADE and M7 ice-nucleation properties in percent for simulations **passive** with Ullrich and Phillips ice nucleation parameterization. If not indicated otherwise domain averages are given. Values are based on vertically-averaged concentrations and correspond to Figures 12 and 13.

|  | Ullrich | Phillips |
|---|---|---|
| INP in mixed-phase clouds (domain average) | -96 | -15 |
| INP in mixed-phase clouds (average over Atlantic) |  | -17 |
| INP in mixed-phase clouds (average over Mediterranean Sea) | -95 | -9 |
| INP in ice-phase clouds (average over Atlantic) | 65 | -39 |
| INP in ice-phase clouds (average over Mediterranean Sea) | - 0 | - 9 |
| INP in ice-phase clouds (average over Africa) | - 0 | -12 |
| frozen solution droplet number (average over Atlantic) | -89 | -20 |
| INP + frozen solution droplet number (domain average where $T < 235\,\mathrm{K}$) |  | -26 |
| solution droplet number in ice-phase clouds (average over Atlantic) | -76 |  |

### 5.3.1 Dust-dominated ice nucleation

As illustrated by the aerosol-surface distribution of dust in mixed- and ice-phase clouds in Figure 11 (b), the average surface of MADE dust particles available for ice nucleation is enhanced as compared to M7. Reasons for this increase are the MADE-only giant dust mode and increased dry deposition of dust from the M7 coarse mode due to its larger $\sigma$ (Section 4.1.2). At comparable number concentrations, the ice nucleation potential increases with the average surface of particles (Phillips et al., 2008) such that INP concentrations tend to be increased for MADE as compared to M7 in clouds in dusty regions (Figures 12, 13, Table 7).

In mixed-phase clouds (Figure 12), M7 INP are reduced by about 15% as compared to MADE for the Phillips and by more than 90% for the Ullrich parameterization. The strong difference for the Ullrich as compared to the Phillips parameterization is a result of a similarly dramatic difference in ice-nucleation-active dust, occurring because all MADE dust but only coated M7 dust is considered for the Ullrich ice nucleation parameterization in mixed-phase clouds (the Phillips parameterization is based on total dust in both cases (Table 2); Figure 11 (b) is thus only relevant for the Phillips parameterization, the corresponding distribution for the Ullrich parameterization is not shown). As the dominance of aqueous-phase sulfate production in M7 strongly restricts condensation and coating, most M7 dust remains uncoated: In the Mediterranean region about 5% is coated and over the Atlantic values up to 50% are reached. The number of INP is thus strongly constrained by the coating efficiency of dust, which depends on the aerosol model and its specific assumptions.

In dust-dominated ice-phase clouds, we distinguish two regions: The high dust concentrations in the southern part of the domain prevent supersaturations high enough for homogeneous freezing of solution droplets such that ice nucleation proceeds purely heterogeneously as indicated by the absence of frozen solution droplets in Fig. 13. For ice-phase clouds over





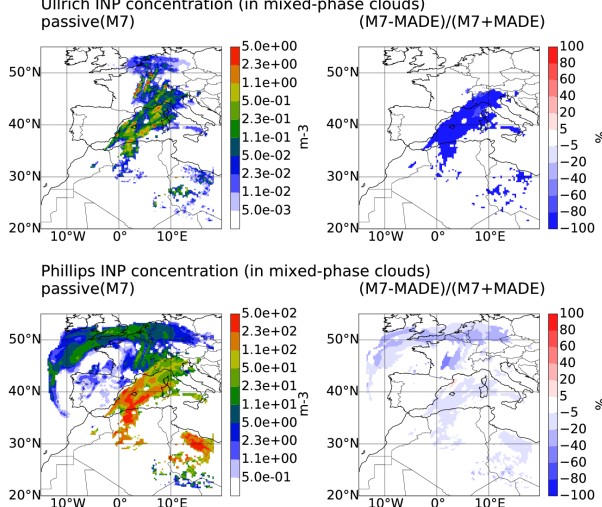

**Figure 12. Mixed-phase INP** concentrations derived from aerosol compositions predicted by M7 (left) and MADE (relative-difference plots on the right) using the Ullrich (first row) and Phillips (second row) parameterizations for **passive** simulations. See Figure 3 for plot details. Contours show vertical averages of concentrations in grid boxes that contain both, ice- and liquid-phase, cloud water.

the Mediterranean Sea, dust concentrations are not high enough to prevent homogeneous freezing completely such that heterogeneous nucleation and homogeneous freezing compete. Due to inefficient coating, the surface distributions of uncoated dust, which is relevant for the Ullrich parameterization, is practically identical to that of total dust, which is relevant for the Phillips parameterization. Difference in Phillips and Ullrich INP are thus both caused by practically identical differences in

ice-nucleation-active dust between MADE and M7. Surprisingly, Phillips INP are reduced by 10% for M7 as compared to MADE and Ullrich INP by less than 1% (Table 7) in both regions. This on the one hand points toward very different relative sensitivities of the Phillips as compared to the Ullrich parameterization. On the other hand, the difference might be influenced by the total INP concentration that the calculation of percentage changes is based on: The Ullrich parameterization results in absolute INP concentrations that are about 2 orders of magnitude higher than for the Phillips parameterization. Similarly

dramatic differences between the Phillips parameterization and an earlier version of the Ullrich parameterization have been discussed by Niemand et al. (2012) for mixed-phase clouds. As a consequence of the low absolute INP concentrations, the Phillips parameterization results in homogeneous freezing being the dominant ice-nucleation mechanism over the Mediterranean Sea, while a competition between homogeneous and heterogeneous freezing occurs for the Ullrich parameterization.

### 5.3.2 Soot-dominated ice nucleation

In the Atlantic part of the domain, soot is the dominant source of INP. The Ullrich parameterization does not consider soot as mixed-phase INP such that mixed-phase Ullrich INP are absent over the Atlantic (Figure 12). For the Phillips parameterization, INP are 17% increased for MADE as compared to M7. This MADE-increase corresponds to an increase in the average surface





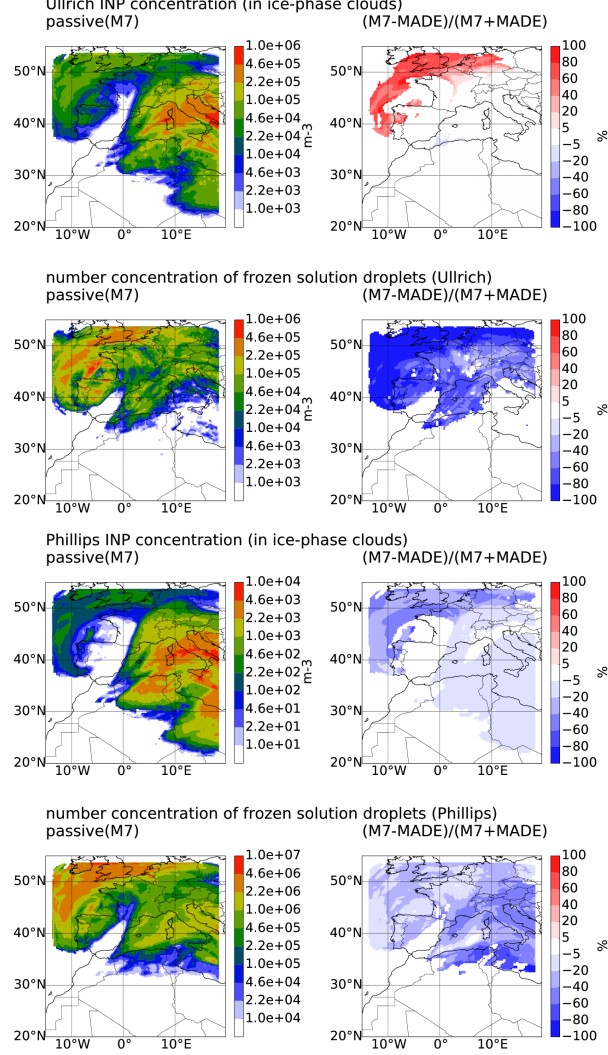

**Figure 13. INP and frozen solution droplet concentrations in ice-phase clouds** derived from aerosol compositions predicted by M7 (left) and MADE (relative-difference plots on the right) using the Ullrich (first and second row) and the Phillips (third and last row) ice nucleation parameterizations. Contours show vertical averages of concentrations in grid boxes that contain ice- but no liquid-phase cloud water. See Figure 3 for plot details.

of MADE soot in comparison to M7 (Figure 11, d). The differences in surface distribution between MADE and M7 are probably a consequence of the distinction between mixed aerosol (modes if and jf in Fig. 1, Tab. 1) and coated soot (modes ic, jc) in MADE while M7 features only a single type of mixed mode (modes ks, as, cs): As discussed for sea salt in Section 4, M7 BC mass is distributed to the large number of all mixed particles, which is interpreted as every mixed-particle having a soot core 5 with a smaller size as compared to soot particles in the insoluble Aitken mode.



Soot-dominated ice nucleation in ice-phase clouds according to the Ullrich parameterization results in a competition between heterogeneous and homogeneous freezing and thus corresponds to the situation of dust over the Mediterranean Sea discussed in the previously. Due to this competition, differences in the number concentrations of INP between MADE and M7 control those of frozen solution droplets for the Ullrich parameterization. Ullrich INP are more than 60% increased for M7 as compared to MADE. This can be attributed to a reduced efficiency of BC coating for M7 in comparison to MADE as illustrated by the surface distributions of uncoated BC in Figure 11 (c). The less efficient coating of M7 BC has two reasons: On the one hand, the SIA burden over the Atlantic is 20% smaller for M7 than for MADE due to MADE nitrate and ammonium. On the other hand, M7 sulfate is primarily produced via aqueous-phase chemistry, which prevents condensation as compared to the gas-phase production pathway of MADE. The increase in M7 Ullrich-INP as compared to MADE shifts the competition between homogeneous and heterogeneous nucleation towards heterogeneous nucleation and leads to a 90% decrease in frozen solution droplets. This mechanisms corresponds to a negative Twomey effect (Kärcher and Lohmann, 2003) with M7 corresponding to polluted and MADE to clean conditions.

For the Phillips parameterization, total INP concentrations are reduced as compared to the Ullrich parameterization such that homogeneous nucleation is the dominant freezing process in the Atlantic part of the domain for this parameterization (Figures 13). Differences between MADE and M7 in terms of INP and frozen solution droplet number concentrations can thus be analyzed separately. A 20% higher abundance of frozen droplets in MADE as compared to M7 reflects differences in the availability of solution droplets for freezing: MADE features a larger number of solution droplets than M7 (+80%, Table 7) because freshly nucleated particles in the MADE Aitken mode are considered for homogeneous nucleation while the corresponding particles in the M7 nucleation mode are excluded (Table 2). Differences in Phillips INP and the corresponding soot distributions between MADE and M7 qualitatively follow the mixed-phase case with about 40% more MADE than M7 INP (Table 7).

### 5.3.3 Comparison of default combinations.

The standard version of COSMO-ART applies the Phillips ice-nucleation parameterization, while the Ullrich parameterization is more suitable for M7 to make use of the simulated difference between coated and uncoated dust. The discussion above shows that the differences between ice-nucleation parameterizations, as drastically illustrated by absolute INP values in ice-phase clouds in dust dominated regions, and in the coupling of the ice-nucleation parameterization to the aerosol scheme, i. e. the consideration of coated vs. uncoated dust and the selection of modes participating in homogeneous freezing, mask the differences between the aerosol schemes. As a consequence, differences between MADE-Phillips and M7-Ullrich are dominated by both, differences in the ice-nucleation parameterization and its coupling to the aerosol scheme.

### 5.4 Buffering effect of cloud microphysics

While the optical properties of aerosols directly influence radiation, their cloud-forming properties as quantified by CCN and INP can affect radiation and precipitation only indirectly via their influence on cloud droplet and ice crystal number. In simulations with aerosol-cloud interactions (simulations **coupled** in Tab. 3), CCN and INP correspond to the activation and ice





nucleation rate and thus to sources of droplet and crystal number. In addition, droplet and crystal number concentrations are subject to cloud microphysical processes, including number sinks like collision-coalescence, riming and aggregation. These processes modify the effect of aerosol differences on droplet and crystal number concentration as compared to their effect on CCN and INP. The mediating effect of cloud microphysics on relative changes $\Delta N/N$ in droplet or crystal number concentra-

tion $N = N_{\text{droplet}}, N_{\text{crystal}}$ that result from a relative changes $\Delta CN/CN$ in cloud nuclei CN=CCN, INP can be quantified by a relative sensitivity or susceptibility (McComiskey et al., 2009; Glassmeier and Lohmann, 2016):

$$s := \frac{\mathrm{d}\ln N}{\mathrm{d}\ln CN} \approx \frac{\Delta N/N}{\Delta CN/CN} \tag{12}$$

These susceptibilities can be determined from double logarithmic plots of $\ln N$ as function of $\ln CN$. The susceptibility is a characteristic of the cloud microphysics scheme and its value will be different for different cloud regimes and states. To get a

sampling of these regimes and states that is representative for the studied case, we make use of the horizontal variability in the domain and use spatially resolved data from simulations with 2-moment microphysics (simulations **coupled** in Tab. 3), temporally and vertically averaged over cloudy grid points in the same way as the data depicted in the contour plots of Figures 10 to 13. As the cloud microphysics scheme that mediates the relationship between $CN$ and $N$ is the same for MADE and M7, we combine data points from simulations with both schemes. The resulting fits are shown in Figure 14, for warm, mixed-phase

and cirrus clouds.

Cloud droplet number concentrations are significantly smaller than predicted concentrations of CCN (Figure 14, a). On the one hand, this arises because our CCN diagnostic does not take into account the competition of different droplets for water vapor, which is considered in the nucleation rate computed in the **coupled** simulations (Sections 2.3,3). On the other hand, growth by collision-coalescence as a droplet sink plays a role in modifying N (a similar argument is given by McComiskey

et al. (2009)): Colors encode the average mass of cloud droplets and rain drops. Small values correspond to recently formed clouds where droplets are too small for efficient collisions. The corresponding data points thus lie closest to the one-to-one line. Large average mass corresponds to raining clouds with efficient collisions and strongly reduced drop number. The coefficient of determination may be interpreted such that 73% of the relative variability in warm cloud droplet number can be explained by relative variability in the available CCN. Differences in the microphysical state of the cloud, for which hydrometeor size is

a proxy, partly account for the unexplained 27% of variance as indicated by the systematic color pattern.

Probably due to ice multiplication by rime-splintering, ice crystal numbers are larger than the number of INP in mixed-phase clouds (Figure 14, b). Ice multiplication as an important source of ice crystal number next to freezing can also explain that crystal numbers are only weakly dependent on INP, which account for only 21% of variance. An additional factor is crystal sedimentation, which provides a number sink that has no analog in warm clouds because cloud droplet sedimentation is

negligible. For a given INP concentration, crystal concentration instead increases with increasing glaciation as defined by the fraction of frozen to total cloud water. Like hydrometeor size in the warm case, this glaciation fraction might be considered a proxy for the state of the cloud.

In cirrus clouds, ice crystal number concentrations tend to be smaller than the sum of INP and frozen solution droplets (Figure 14, c), likely as a result of sedimentation as an ice crystal sink. Sedimentation is most effective for large hydrometeors in



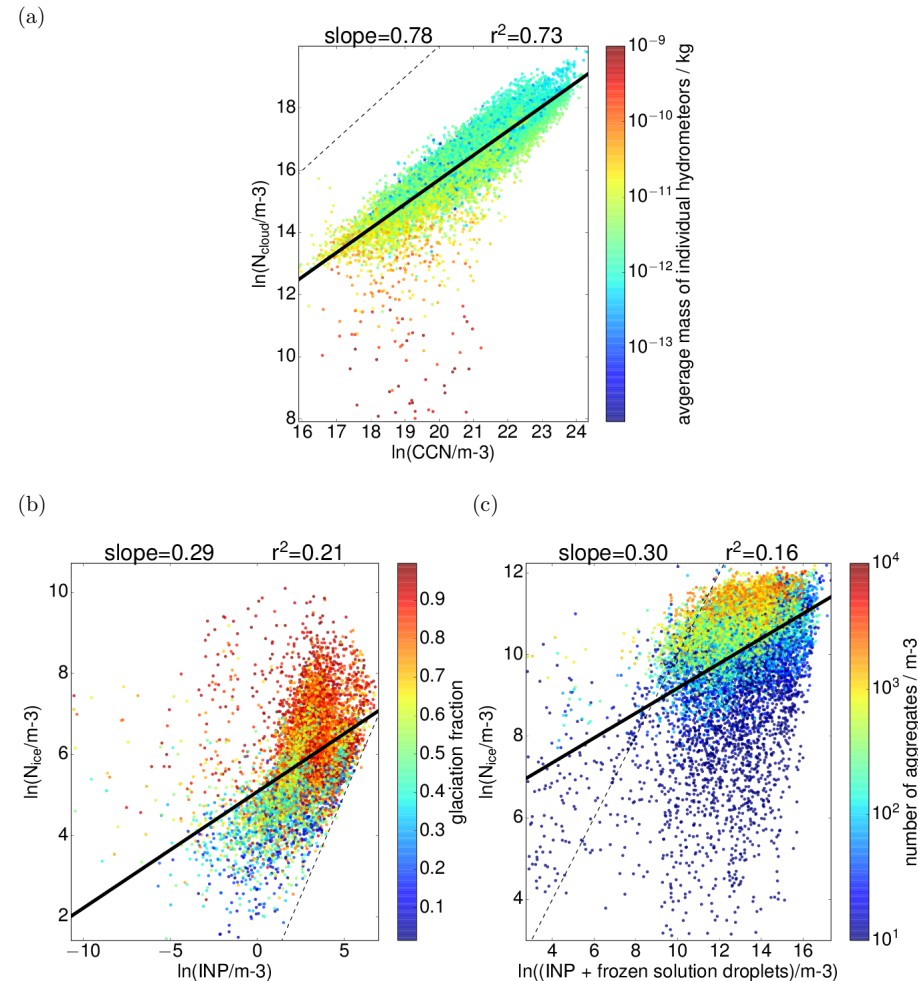

**Figure 14. Relationship between CCN/INP and cloud droplet number concentration $N_{cl}$ and ice crystal number concentration $N_{ci}$.**
The figures show double logarithmic scatter plots of the cloud droplet number concentration $N_{cl}$ as function of CCN concentration in liquid clouds (a) and the ice crystal number concentration $N_{ci}$ as function of INP concentration in mixed-phase clouds (b) and as function of the combined concentration of INP + frozen solution droplets in cirrus clouds (c). Warm and mixed-phase clouds are defined as in Figures 10 and 12, cirrus clouds are restricted to regions with competition of heterogeneous and homogeneous freezing as indicated by non-vanishing numbers of both, INP and solution droplets. Data points represent temporally and vertically averaged values from **coupled** simulations with the Phillips ice nucleation parameterization at every grid point (see text for details). Only every 5th data point used for the fit is displayed. Solid black lines illustrate least-square fits with slope and coefficient of determination $r^2$ as indicated in plot titles. All fits are significant. Black dashes indicate one-to-one lines. Colors denote the mass of individual hydrometeors, averaged over all cloud droplets and rain drops (a), the glaciation fraction, i. e. the ratio of frozen to total cloud water (b), and the number of ice-crystal aggregates (c).

the snow category of the microphysics scheme that result from the aggregation of individual ice crystals. Although aggregation is not very efficient at the low temperatures of cirrus clouds, the degree of aggregation seems a possible proxy for the micro-




**Table 8. Relative difference in cloud droplet and ice crystal number $N$ between MADE and M7** as predicted according to Equation 13 from fitted susceptibilities $s$ and domain-averaged values of relative differences in cloud nuclei $\Delta CN/CN$ from simulation **passive** (Table 7).

| cloud | $\Delta CN/CN$ | $s$ | $\Delta N/N$ |
|---|---|---|---|
| liquid | -54% | 0.78 | -42% |
| mixed-phase | -15% | 0.29 | -4% |
| cirrus | -26% | 0.30 | -8% |

physical cloud state: The color coding based on the number of aggregates in Figure 14 (c) can explain variance in addition to differences in the number of INP and frozen droplets, which explain 16%. Data points above the one-to-one line are probably related to the homogeneous freezing of cloud droplets that are advected to regions with temperatures colder than 235 K. This freezing of cloud droplets has to be distinguished from the homogeneous freezing of solution droplets predicted by the

ice-nucleation parameterization. Homogeneous freezing of cloud droplets is restricted to cirrus or ice-phase clouds with liquid origin, e. g. outflows from convective clouds or high-reaching tops of nimbostratus clouds. It seems that this cirrus regime contributes to the variability of the relationship between INP and frozen solution droplets and ice crystal number concentrations in the regime of low crystal concentrations.

Our discussion of Figure 14 illustrates that a detailed understanding of the influence of cloud microphysics on the coupling

between parameterized cloud nuclei and the number concentration of cloud droplets and ice crystals requires a more detailed analysis, including a separation of cloud regimes and microphysical cloud states. This is beyond the scope of the current study. By rearranging equation 12 according to

$$\frac{\Delta N}{N} \approx s \cdot \frac{\Delta CN}{CN} \qquad (13)$$

the values of the fitted slopes with $s < 1$ nevertheless show that cloud microphysics dampens, or buffers, the effect of differ-

ences in the aerosol representation, i.e. MADE vs. M7, on $N$ as compared to $CN$: An aerosol signal in $CN$ will overestimate the signal in $N$ and thus an effect on clouds (Table 8). The buffering effect on the ice phase is stronger than in liquid clouds. Nevertheless, Table 8 shows that the details of the chemistry and aerosol scheme have a non-vanishing effect on all three cloud types investigated.

Note that we prefer the susceptibility-based estimate over a direct comparison of $N$ for coupled simulations with MADE

and M7 because coupled simulations do not have identical meteorologies for the different aerosol schemes. Differences in meteorology, specifically in supersaturation and temperature, influence differences in $CN$ in addition to aerosol differences. Not taking this additional meteorological variability into account would result in an overestimation of $\Delta CN/CN$ and consequently of $\Delta N/N$.





## 6  Conclusions

We have coupled the M7 aerosol scheme (Vignati et al., 2004) and the computationally efficient sulfur chemistry of Feichter et al. (1996) with the regional aerosol- and reactive trace gas model COSMO-ART with interactive meteorology (Vogel et al., 2009). While the M7 aerosol framework was designed for climate applications, the full gas-phase chemistry and the aerosol scheme MADE in COSMO-ART emerged from regional-scale air-quality applications. The availability of the two different descriptions of aerosol and aerosol-related chemistry within the same modeling framework allows for a detailed comparison and process-level understanding of their differences. As both aerosol schemes adopt a modal 2-moment approach, this comparison especially reveals the uncertainty in aerosol modeling arising from design and parameter choices within this framework. Here, we have compared the aerosol modules in a case study of a Saharan dust outbreak reaching Europe.

In a sensitivity study with identical emissions and identical parameterizations of dry and wet deposition for both schemes, we identified the following sensitivities of simulated atmospheric aerosol burden, sorted in order of decreasing importance:

1. chemical reactions considered for sulfate production (70% difference, Figure 6, Table 4)

2. coarse-mode composition (60% difference, affecting the sulfate burden in Figure 4)

3. modal standard deviation (10-20% difference in dust/sea salt size distributions in Figure 5)

4. accumulation and Aitken mode composition (5% difference in POA and BC burden in Figure 3)

5. oxidant fields for sulfate production (1% difference, Table 5)

The strong sensitivity of the aerosol burdens to sources is well recognized by the aerosol modeling community (Mann et al., 2014). Our example especially stresses that uncertainties are not limited to prescribed anthropogenic emissions but extend to parameterized aerosol sources like chemically derived sulfate (1.). Aerosol sink processes are similarly sensitive and strongly increased for large internally-mixed aerosol particles (2.). Modal standard deviation is an inevitable, but important parameter of a 2-moment scheme, especially for the dry deposition and impaction scavenging of coarse mode particles (3.). Aitken and accumulation mode aerosol mass is less affected by dry deposition and impaction scavenging such that their composition and standard deviation is less important in determining aerosol burden (4.).

In contrast to these sensitivities, we find that climatological oxidant fields perform as well as hourly values (5.). In the investigated case, emissions of $SO_2$ are largely restricted to cloudy regions such that sulfate is predominantly produced by aqueous-phase chemistry. Also, the effect of opposing deviations of the climatological oxidant fields from the hourly values compensate in their effect on the overall aqueous-phase reaction rate. Although our result is not generally applicable, it hints to low sensitivities of the sulfate burden to oxidant fields at least in some cases and confirms the validity of a constant-oxidant-field approach for the efficient aerosol-related chemistry. It has to be explored in future research if extensions of the approach to other secondary aerosols, namely SOA, nitrate and ammonium can provide a sufficiently accurate and computationally feasible way to account for these species in climate applications.





A comparison of both aerosol schemes in their default setups is strongly influenced by those aerosol species that are only considered by MADE, namely nitrate, ammonium, SOA and unspeciated aerosol and by the likewise MADE-specific giant modes, especially for dust. We find that the additional sulfate produced by the M7-aqueous chemistry partially compensates for the additional nitrate and ammonium aerosol specific to MADE.

The additional MADE species play a large role for the sensitivities of CCN and optical properties to the aerosol scheme. MADE features 20% (mixed-phase clouds) to 50% (liquid clouds) higher CCN concentrations than M7 due to MADE-specific soluble species, i.e. nitrate, ammonium, SOA and unspeciated aerosol. M7-specific aqueous-phase-derived sulfate mainly increases already CCN-sized particles and hardly affects the M7 CCN concentration. MADE-AOD is about 50% increased as compared to M7 in anthropogenically influenced regions. Over dust-dominated African regions, MADE-AOD is 10% larger

than M7-AOD, partly due to the MADE-only giant dust mode. Differences in the parameterizations of aerosol optical properties between MADE and M7 are found to be less important than the differences in burden.

The INP potential of an aerosol depends on its surface area. For the Phillips ice-nucleation parameterization, which is independent of the mixing state and coating of an aerosol, we find that differences in dust and soot burden and surface area explain differences in INP. For the Ullrich parameterization, which depends on the coating state of soot and dust, the abundance

of secondary inorganic aerosol available for coating becomes more important in explaining differences in INP numbers than the burden and size of ice-nucleation active species. In conditions where ice nucleation is dominated by homogeneous freezing of solution droplets, ice crystal concentrations are influenced by the number of soluble aerosol particles available for homogeneous freezing. As the total aerosol number is dominated by freshly nucleated particles, we find the minimum size of particles considered large enough to freeze homogeneously to be a relevant parameter. Large differences between the Phillips and

Ullrich ice-nucleation parameterizations show, however, that uncertainties in parameterizing ice nucleation, in terms of the ice-nucleation spectrum as well as concerning the choice of inputs from the aerosol scheme, are more important than uncertainties in modeled aerosol number, amount and composition.

Applying a susceptibility-based approach, we find that cloud microphysics dampens the differences in CCN and INP arising from differences between MADE and M7 aerosol microphysics along the line of clouds as buffered systems (Stevens and

Feingold, 2009). The effect is especially pronounced for the ice phase. Nevertheless, both schemes result in significantly different cloud droplet and ice crystal number concentrations. Uncertainties in representing aerosol and aerosol processes thus carry over not only to the direct optical properties of aerosol but also to the representation of clouds. For a propagation of this signal to precipitation, however, further buffering effects are expected (Glassmeier and Lohmann, 2016).

In summary, differences between the two aerosol microphysics schemes and resulting differences in radiative properties and

aerosol-cloud interactions originate mainly in different structural assumptions of the schemes, in particular concerning aerosol species, chemical reactions, modal composition and standard deviation, and inputs for the ice nucleation parameterization. Resulting impacts on radiative properties and aerosol-cloud interactions are buffered: On the one hand by compensating structural differences between additional sulfate from aqueous-phase chemistry for M7 and additional nitrate, SOA und unspeciated aerosol for MADE. On the other hand by sublinear relationships between aerosols and clouds.





We conclude that the new model version COSMO-ART-M7 simulates satisfying aerosol burdens in comparison to the established and observationally validated modeling framework COSMO-ART (Knote et al., 2011). Differences in burdens can be attributed to the choice of uncertain parameters, in particular modal standard deviation, and different structural assumptions in the form of missing species like SOA, nitrate and ammonium, and the focus of M7 design on climate applications,

which stresses the difference between soluble and insoluble modes but only considers one type of mixed aerosol. Ice nucleation not only depends on the mixing state of dust but also on an accurate representation of the dust surface. The latter is lost for internally mixed dust and soot. This raises the question if the representation of dust surfaces in M7 should be improved by following MADE in excluding dust from the mixed-modes and adding a separate coated dust mode. To keep the original number of modes and the corresponding computational costs the same, the uncoated accumulation and coarse dust modes

could be replaced by a coated and an uncoated dust mode of intermediate size. Additionally, the consequences of different approaches to mode reorganization and the description of particle growth by coating between MADE and M7 should be further investigated. For MADE and air quality applications, an aqueous-phase chemistry that is efficient enough for the standard setup (a detailed aqueous-phase chemistry and wet-scavenging scheme for COSMO-ART has been developed by Knote and Brunner (2013)), seems a relevant objective of future model development. Additionally, nucleation scavenging and CCN/INP

depletion will affect modeled MADE and M7 aerosol as well as cloud properties and its implementation is part of ongoing model development.

## 7   Code availability

Model code is subject to licensing following http://www.cosmo-model.org/content/consortium/licencing.htm for COSMO-ART and additionally https://redmine.hammoz.ethz.ch/projects/hammoz/wiki/1_Licencing_conditions for COSMO-ART-M7.

Licences are free of charge for research applications. They are available from the authors upon request.

## 8   Data availability

Simulation output is archived on ETH Zurich infrastructure and available from the authors upon request.

*Competing interests.*   BV is a co-editor of ACP. Besides, the authors declare that they have no conflict of interest.

*Acknowledgements.*   We thank Grazia Frontoso for isolating an M7 boxmodel from ECHAM-HAM, Romy Ullrich for sharing the imple-
mentation of her ice-nucleation parameterization and Ulrich Blahak and Axel Seifert for providing their cloud microphysics scheme. Max Bangert is gratefully acknowledged for his assistance, especially with the coupling of M7 to ART and the initial adaptation of the aerosol-cloud and aerosol-radiation interactions to M7. We also thank Isabel Kraut for her introduction to the application of KPP in COSMO-ART. Jianxiong Cheng and Olga Henneberg are acknowledged for first versions of wet chemistry and an M7-adapted impaction scavenging routine. We further thank Johann Feichter for clarifying some details of his chemistry scheme, Tanja Stanelle for discussions about the optical



properties of dust in ART and Annette Miltenberger for her opinion on the choice of the simulated case. Isabelle Bey is acknowledged for her support in developing COSMO-ART-M7. FG and AP were funded by the ETH-domain CCES project OPTIWARES (41-02).

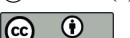


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
