# Peer review of "A comparison of two chemistry and aerosol schemes on the regional scale and resulting impact on radiative properties and liquid- and ice-phase aerosol-cloud interactions"

_Atmospheric Chemistry and Physics, 2016_

## Referee Comment (RC1) · Anonymous Referee #2 · 6 Mar 2017

The authors compared impacts of two different aerosol module, MADE with full gas-phase chemistry and M7 with a constant-oxidant-field-based sulfur cycle, on aerosol burden, as well as aerosol-cloud interactions. They found that aqueous-phase sulfate production, the selection of aerosol species and modes and modal composition are more important than parametric choices for aerosol populations. Differences in cloud droplet and ice crystal number concentrations are buffered by cloud microphysics. This study could improve the understanding on implication of aerosol schemes in air quality and climate models. Before this manuscript can be considered for publication, I have a few comments that need to be addressed by the authors.

[Figure]

Major comment:

The author compared MADE designed for air quality applications and M7 for climate projections in this study and concluded the importance of different processes or parameterization. However, one of the interesting information is not clearly presented and should be discussed. That is which processes are most important for air quality models and which processes are important for climate models, and possible improvement of future models based on your results.

Other minor comments:

Page 5: I suggest the authors to add a table to compare the detail of aerosol processes between these two aerosol modules.

Page 7 Line 3: Change 2.2.1 to 2.3, because Aerosol-radiation interactions are not relevant to 2.2 Sulfur chemistry.

Page 8 Line 23: Why the author chose a Saharan dust outbreak reaching Europe in May 2008?

Table: MADE passive simulation aqueous-phase chemistry and climatological oxidant fields. It may confuse readers if it shows 'y' as the same as M7. The authors should clarify it in table caption.

The authors used $(f1 - f2) / (f1 + f2)$ to quantify relative differences in this study. I think the it should be $(f1 - f2) / [(f1 + f2)/2]$ instead.

Page 19 Line 10: It may not be valuable to compare secondary inorganic aerosol between MADE and M7. Although M7 only simulate sulfate, the particle is actually $(NH4)2SO4$, $NH4HSO4$ in the atmosphere. Many climate model consider sulfate mass as $NH4HSO4$ instead of $SO4$. The author did not describe how is sulfate mass treated in M7.

Page 28 Line 13: Because the authors are comparing two aerosol modules, it is better

to show the figures for both of the two modules. Do they have the same information with the combined data?

Page 28 Line 16: Change smaller to lower.

Page 31 Line 13: Please clarify these conclusions are probably region-dependent. As I know, the aqueous oxidation of sulfate is sensitive over Europe, but some other regions are not. Also, please change chemical reactions to aqueous oxidation, because chemical reactions may mislead readers.

---

## Referee Comment (RC2) · Anonymous Referee #1 · 1 Apr 2017

This paper compares two very different aerosol modules embedded within the same model framework. They first explored the impact of parameters/oxidant fields that may affect the aerosol formation and distribution and then they compared the impact of the two modules on AOD, CCN, INP and formed cloud droplets/ice numbers. The final main conclusion is the structural differences between the two models (like chemical reaction schemes, species amount) play the major role in determining the aerosol production or burden, which is well expected. Another main conclusion is that the differences of aerosol burdens and nucleation schemes have impact on the formed cloud droplet and ice particles numbers but the impact is diminished or buffered by the cloud

microphysics.

Overall, the paper is well written and the study is thorough and contributes to the understanding of aerosol formation and their impact on clouds. Yet it is not clear in many places, I recommend the authors address my comments below before it can be accepted for publication.

General comments: 1) Using (f1-f2)/(f1+f2) to see the percentage changes can be confusing and misleading, especially when f1 and f2 are quite different. Suggest show both fields and then show the percentage differences. 2) I also suggest the authors add an acronym list at the end of the paper. 3) I also suggest the authors show the precipitation field from the simulation.

Specific comments/questions: Page 3, show full name of MADE

Page 4, Fig 1, caption. Show the meaning of the 2-letter abbreviations in the caption and add an acronym appendix.

Page 4, line 13. If there is not nucleation scavenge, does impact scavenge take the hygroscopicity of aerosols into consideration? In another way, can increased SO4 coating affect the wet deposition of non-sulfate aerosols?

page 5, line 14, how many insoluble modes in M7? seems 3 in Fig. 1.

Page 7, How a PDF of updraft velocity is used in the activation scheme? Please add a few sentences to explain it.

Page 8, line 20. Add "excess" in front of water vapor.

Page 11, Line 7/8, why use "ca." instead of "$\sim$" which is more straightforward?

Page 11, last line, is the impaction scavenge scheme in the models able to describe such effect? If so, please add a few line describing the impaction scavenge scheme in the previous section.

Page 12, Line6, please clarify how you calculate the size distribution of internally mixed mode (like as,cs in M7) and how they are used to calculate the distributions of each species showed in Fig.5.

Page 12, line 16. From table 4, sea salt in M7 is 12% more (or ∼24% actually). Is this the mainly reason rather than being smaller?

Page 13, Figure 3. Why use (f1-f2)/(f1+f2) instead of ((f1-f2)/(f1+f2)*0.5) to show the percentage changes. By using (f1-f2)/(f1+f2), you are essentially diminishing the percentage changes by a factor of 2 in all your discussions.

Page 14. Does BC in accumulation mode in M7 have more SO4 coating? If so, this may explain less BC in this mode via wet deposition.

Page 14, last line. does SO4 coating affect the impact scavenge? Is there nucleation removal or other removal mechanism beside this?

Page 15, line 2. Switch "4,3". What is SIA?

Page 16, fig 5. What is the "Total distribution" here, which curve? How are the distributions of each species calculated is not clear from the caption.

Page 19, Line 5. Is SO4 coating from aqueous production of SO4 assigned to accumulation mode only or mainly? how is this determined?

Page 19, last 4 lines. What is the purpose of this compensation of sulfate burden in MADE ? Is it simply because M7 has more sulfate and MADE has more species? Need to justify why you compare it this way.

Page 28, line 5. Suggest change CN=CCN, INP and N=.... Make it clearer. Also how is the CCN calculated? Use certain S or use the w* mentioned on page 10?

Page 28, 26, "riming-splintering" has very narrow temperature range to apply. Is it possible to show the ice formation from this mechanism to the INP?

Page 29, last line. Please explain why larger aggregation number actually makes it closer to the 1:1 line.

**ACPD**

---

## Author Comment (AC1) · 3 Jun 2017

We thank referee 1 for the constructive and detailed comments and address each of them and corresponding changes to the manuscript below. Additional editorial changes are documented in a separately provided manuscript version with marked-up differences.

**General comments:**

[Figure]

1. Using (f1-f2)/(f1+f2) to see the percentage changes can be confusing and mis-leading, especially when f1 and f2 are quite different.

   *In the submitted version of this manuscript we had defined the percentage change as $\Delta_{old} = (f_1 - f_2)/(f_1 + f_2)$ because this expression has a nicer behavior for large relative differences than the version favored by the reviewer, $\Delta_{new} := 2\Delta_{old}$. For $f_1 >> f_2 \Rightarrow \Delta_{old} = 100\%$ such that $-100\% < \Delta_{old} < 100\%$ whereas $-200\% < \Delta_{new} < 200\%$ and $\Delta_{new} = 100\% \Leftrightarrow f_1 = 3f_2$. Despite this we agree with the reviewer that for smaller percentage differences $\Delta_{new}$ may be more intuitive and since both referees prefer $\Delta_{new}$ we have revised the manuscript accordingly.*

2. Suggest show both fields and then show the percentage differences.

   *During our analysis we did not find distinct differences in the spatial patterns of the compared fields. Therefore additional panels showing the second field do not add additional insight as is illustrated in Fig. 1 below, which shows Fig. 3 from the manuscript with the MADE field included. As this can also be inferred by the absence of spatial structure seen in the difference plots, we decided not to include additional panels for the sake of clarity as the manuscript already contains a lot of figures.*

3. I also suggest the authors add an acronym list at the end of the paper.

   *A list of acronyms is added.*

4. I also suggest the authors show the precipitation field from the simulation.

   *As simulations with aerosol-cloud interactions are not meteorologically nudged simulations we deliberately abstained from showing precipitation fields. In order to attribute changes in precipitation to changes in aerosol treatment one would have to perform an ensemble of simulations in order to first capture the natural variability of the system and then to test if significant differences are found in the*

*precipitation field exceeding the natural variability. Although this analysis would certainly be interesting it is beyond the scope of this study. The authors felt that showing difference plots in precipitation of the MADE and M7 simulations in comparison could be misleading. Such a plot may suggest to the reader that these changes could be attributed to the two different microphysics scheme, which is not necessarily the case.*

**Specific comments:**

- Page 3, show full name of MADE:

  *Changed accordingly.*

- Page 4, Fig 1, caption. Show the meaning of the 2-letter abbreviations in the caption and add an acronym appendix.

  *Changed accordingly.*

- Page 4, line 13. If there is not nucleation scavenge, does impact scavenge take the hygroscopicity of aerosols into consideration? In another way, can increased SO4 coating affect the wet deposition of non-sulfate aerosols?

  *page 5, line 13 has been changed for clarification:*

  "The parameterization is applied to the wet aerosol radius such that the hygroscopicity of an aerosol particle may affect is scavenging efficiency by impaction."

- page 5, line 14, how many insoluble modes in M7? seems 3 in Fig. 1

  *Yes, there are only 3 insoluble modes. Changed accordingly.*

- Page 7, How a PDF of updraft velocity is used in the activation scheme? Please add a few sentences to explain it.

*page 7, line 17 has been extended for clarification:*

"To take into account the sub-gridscale updraft variability, the number concentration of activated aerosol particles is determined by numerically averaging over a Gaussian probability density function (PDF) of updraft velocities about the grid mean value rather than using the number concentration of particles that are activated for the grid-mean updraft. The standard deviation of the PDF depends on the turbulent kinetic energy."

- Page 8, line 20. Add "excess" in front of water vapor.

  *The expression "excess water vapor" might be interpreted such that a saturation adjustment technique for the ice phase would be applied. This is not the case. The line has therefore been clarified in the following way:*

  "[Ice-nucleation in ice-phase clouds follows the previous approach of MADE and] converts water vapor into ice."

- Page 11, Line 7/8, why use "ca." instead of "$\sim$" which is more straightforward?

  *Changed accordingly.*

- Page 11, last line, is the impaction scavenge scheme in the models able to describe such effect? If so, please add a few line describing the impaction scavenge scheme in the previous section.

  *page 5, line 13 has been changed for clarification:*

  "The description of impaction scavenging is based on an aerosol- and hydrometeor-size dependent collection efficiency. It considers inertial impaction and impaction from Brownian diffusion and interception but not phoretic effects."

- Page 12, Line6, please clarify how you calculate the size distribution of internally mixed mode (like as,cs in M7) and how they are used to calculate the distributions of each species showed in Fig.5.

Page 16, fig 5. What is the "Total distribution" here, which curve? How are the distributions of each species calculated is not clear from the caption.

*The caption of Fig. 5 has been clarified in the following way:*

"Species include sea salt (SS), dust (DU), sulfate ($SO_4$), soot (BC) and primary organic carbon (POA). Individual lognormal modes are determined from the vertical sum and horizontal average of the the corresponding dry masses and numbers. The full, multimodal distribution emerges as the sum of individual log-normal modes. For mixed modes, lognormal modes of individual species are obtained by weighting the mixed-modal distribution by the fraction that the respective species contributes to the total mass in the mixed mode."

- Page 12, line 16. From table 4, sea salt in M7 is 12% more (or ∼24% actually). Is this the mainly reason rather than being smaller?

  *Yes, the text has been adapted accordingly:*

  "[The M7 coarse-mode coating could be more effective because] sea salt is more abundant and particles are smaller such that a larger surface for condensation is available."

- Page 13, Figure 3. Why use (f1-f2)/(f1+f2) instead of ((f1-f2)/(f1+f2)*0.5) to show the percentage changes. By using (f1-f2)/(f1+f2), you are essentially diminishing the percentage changes by a factor of 2 in all your discussions.

  *See general comment (1).*

- Page 14. Does BC in accumulation mode in M7 have more SO4 coating? If so, this may explain less BC in this mode via wet deposition.

  *If the mixed M7 accumulation mode (as), which contains BC coated by sulfate, were to be scavenged more efficiently than the mixed MADE mode containing a soot core (jc), we would expect a decrease in M7 POA as well as BC, because*

[Figure]

*these two species are internally mixed and thus always scavenged together. As we do not find this to be the case, we conclude that differences in wet deposition alone thus cannot offer an explanation for the differences in BC.*

- Page 14, last line. does SO4 coating affect the impact scavenge?

*The current argument in the manuscript successfully explains differences in aerosol abundance between MADE and M7 based on differences in their dry radii and corresponding differences in deposition. The reviewer suggests that this argument might be incomplete because differences in the amount of sulfate coating will lead to differences in water uptake such that the wet radii may differ in a different fashion than the dry radii. We do not think that this the case for the following reasons: The M7 Aitken mode contains less SO4 than MADE, the M7 accumulation mode also contains less SO4 but more sea salt, the M7 coarse mode contains more sea salt and more SO4. For BC, the decreased SO4 coating strengthens our argument based on the dry aerosol radii instead of using the wet radii because it means even smaller particles due to decreased water uptake. For SS, we can assume that the SO4 coating is of minor importance as compared to the overall water uptake because sea salt is more hygroscopic than sulfate (kappa value of 1.12 as compared to 0.6 for sulfate). For POA, we can likewise assume that an increased water uptake due to the increased sea salt strengthens our argumentation made for the dry radius. Thus, although the amount of hygroscopic aerosol (sea salt and SO4) will control wet radii and deposition in an absolute terms, differences in dry radii seem to be a sufficient proxy for differences in wet radii, at least in our specific cases.*

- Page 14, last line. Is there nucleation removal or other removal mechanism beside this?

*Nucleation scavenging is not considered in this study so that there are no other wet removal mechanisms.*

- Page 15, line 2. Switch "4,3". What is SIA?

  *Changed accordingly and introduced meaning of SIA.*

- Page 19, Line 5. Is SO4 coating from aqueous production of SO4 assigned to accumulation mode only or mainly? how is this determined?

  *For clarification the following has been added to page 7, line 2:*

  "SO4(aq) resulting from the aqueous-phase reaction is in most cases assigned to the mixed accumulation mode (mode as in Figure 1 and Table 1) and in fewer cases to the mixed coarse mode (mode cs). This is implemented by a number-based partitioning that favors the more numerous accumulation mode."

  *And the following to page 19, line 5:*

  "[the aqueous-phase chemistry deposits sulfate mainly into the accumulation and to a lesser extent into the coarse mode] (the partitioning between these two modes is based on number and thus favors the more numerous accumulation mode)"

- Page 19, last 4 lines. What is the purpose of this compensation of sulfate burden in MADE ? Is it simply because M7 has more sulfate and MADE has more species? Need to justify why you compare it this way.

  *For clarification, page 19, lines 7-10 have been rephrased as follows:*

  "As discussed, M7 aqueous chemistry produces much higher sulfate concentrations, while MADE features nitrate and ammonium as addition aerosol species. Similar to different secondary organic aerosol (SOA) species, which are often lumped together, we combine sulfate, nitrate and ammonium into a secondary inorganic aerosol (SIA) class, to obtain a quantity that can be compared between MADE and M7. Note that for M7, SIA is identical to sulfate aerosol. From this perspective, the higher contribution of M7 sulfate to the total aerosol burden is compensated for by MADE nitrate and ammonium"

- Page 28, line 5. Suggest change CN=CCN, INP and N=. . .. Make it clearer.

  *The text has been clarified in the following way:*

  "The mediating effect of cloud microphysics on relative changes $\Delta N/N$ in a hydrometeor number concentration $N$ that result from a relative change $\Delta CN/CN$ in the concentration of a cloud nuclei can be quantified [...] This universal equation quantifies liquid-phase (ice-phase) aerosol-cloud interactions when applying it to the droplet (ice crystal) number concentration $N_{\mathrm{droplet}}$ ($N_{\mathrm{crystal}}$) and CCN (INP) by substituting $N = N_{\mathrm{droplet}}$ ($N = N_{\mathrm{crystal}}$) and CN=CCN (CN=INP)."

- Page 28, line 5. Also how is the CCN calculated? Use certain S or use the w* mentioned on page 10?

  *page 10, line 21 has been amended for clarity:*

  "In simulations coupled, the same updraft parameterization as used in simulations passive (i.e. no PDF) is applied for the online as well as offline calculation of CCN."

- Page 28, 26, "riming-splintering" has very narrow temperature range to apply. Is it possible to show the ice formation from this mechanism to the INP?

  *Ice multiplication does indeed not seem to be the dominant mechanism to explain the high ice crystal concentrations as Fig. 2 below does not show a trend with temperature. To reflect this, the text has been adapted as follows:*

  "Ice crystal number concentrations are always higher than the number concentration of INP in mixed-phase clouds (Figure 14, b). This can be attributed to ice crystal sources other than the heterogeneous freezing of cloud droplets. In our model, these are the freezing of rain drops, ice multiplication by rime-splintering and the sedimentation of ice crystals from aloft. These INP-independent ice crystal source processes [can also explain that crystal numbers are only weakly dependent on INP]"

- Page 29, last line. Please explain why larger aggregation number actually makes it closer to the 1:1 line

  *We cannot identify a trend that larger aggregation numbers are closer to the dashed 1:1 line. They instead seem to increase with ice crystal concentration, i.e. along the y-axis of the plot.*
* * *
[Figure]

**Fig. 1.** Supplementary plot for Fig. 3: M7, MADE and percentage difference fields.

**Fig. 2.** Supplementary plot for Fig. 14: Role of temperature.

---

## Author Comment (AC2) · 3 Jun 2017

We thank referee 2 for the constructive and detailed comments and address each of them and corresponding changes to the manuscript below. Additional editorial changes are documented in a separately provided manuscript version with marked-up differences.

**Major comment:**

The author compared MADE designed for air quality applications and M7 for climate

projections in this study and concluded the importance of different processes or parameterization. However, one of the interesting information is not clearly presented and should be discussed. That is which processes are most important for air quality models and which processes are important for climate models, and possible improvement of future models based on your results.

*We agree with the reviewer that this distinction is necessary for the clarity and direction of the manuscript. This discussion is now included in the revised manuscript on page 33, line 1ff as follows:*

"We conclude that the new model version COSMO-ART-M7 simulates satisfying aerosol burdens in comparison to the established and observationally validated modeling framework COSMO-ART (Knote et al., 2011). Differences in burdens can be attributed to the choice of uncertain parameters, in particular modal standard deviation, and different structural assumptions in the form of missing species like SOA, nitrate and ammonium, and the choice of modes in terms of solubility and mixing state. This study provides the opportunity to discuss these choices in terms of the air-quality and climate objectives they are designed for. For climate applications, a computationally efficient aerosol scheme, such as M7, is needed that permits an as realistic as possible computation of radiative effects and aerosol-cloud interactions. As discussed earlier, simplified chemistry seems a viable option to save computational cost. In terms of aerosol-cloud interactions, the M7 approach to distinguish soluble from insoluble aerosol but to only consider one mixing state might be biased towards warm clouds. Ice nucleation not only depends on the mixing state of dust but also on an accurate representation of the dust surface. The latter is lost for internally mixed dust and soot. This raises the question if the representation of dust surfaces in M7 should be improved by following MADE in excluding dust from the mixed-modes and adding a separate, coated dust mode. To keep the original number of modes and the corresponding computational costs the same, the uncoated accumulation and coarse dust modes could be replaced by a coated and an uncoated dust mode of intermediate

[Figure]

size. When applying MADE for air-quality applications, the chemical speciation as well as the abundance of individual aerosol species, precursor gases and pollutant gases such as tropospheric ozone are of great interest. For this purpose, an as simplified treatment of chemistry as is used in M7 is no longer justified and a far more complex treatment is needed. However, this study confirms that an aqueous-phase chemistry that is efficient enough for the standard setup (a detailed aqueous-phase chemistry and wet-scavenging scheme for COSMO-ART has been developed by Knote and Brunner (2013)), may be a relevant objective for future model development."

**Minor comments:**

- Page 5: I suggest the authors to add a table to compare the detail of aerosol processes between these two aerosol modules.

  *A new table (Tab. 2) has been added.*

- Page 7 Line 3: Change 2.2.1 to 2.3, because Aerosol-radiation interactions are not relevant to 2.2 Sulfur chemistry.

  *Changed accordingly.*

- Page 8 Line 23: Why the author chose a Saharan dust outbreak reaching Europe in May 2008?

  *Desert dust is one of the most effective natural INP currently known. Following Bangert et al (2012), we thus choose a dust outbreak to obtain a significant amount of ice-nucleation active aerosol so that we can analyze the difference between the two aerosol schemes not only in liquid clouds, but also in ice- and mixed-phase clouds. The following text is added to the manuscript for clarification:*

  "Following Bangert et al. (2012), we choose a dust event to ensure sufficient INP concentrations inside our simulation domain in order to compare the implications

of aerosol schemes not only on liquid-phase processes, but also on ice nucleation rates in mixed-phase and ice clouds."

- Table: MADE passive simulation aqueous-phase chemistry and climatological oxidant fields. It may confuse readers if it shows 'y' as the same as M7. The authors should clarify it in table caption.

  *The caption has been clarified as follows:*

  "[A 'y' shows that a model feature is active,] if applicable (aqueous-phase chemistry and climatological oxidant fields are only active for M7 simulations and giant modes only apply to MADE simulations)."

- The authors used (f1 − f2) / (f1 + f2) to quantify relative differences in this study. I think the it should be (f1 − f2) / [(f1 + f2)/2] instead.

  *In the submitted version of this manuscript we had defined the percentage change as $\Delta_{old} = (f_1 - f_2)/(f_1 + f_2)$ because this expression has a nicer behavior for large relative differences than the version favored by the reviewer, $\Delta_{new} := 2\Delta_{old}$. For $f_1 >> f_2 \Rightarrow \Delta_{old} = 100\%$ such that $-100\% < \Delta_{old} < 100\%$ whereas $-200\% < \Delta_{new} < 200\%$ and $\Delta_{new} = 100\% \Leftrightarrow f_1 = 3f_2$. Despite this we agree with the reviewer that for smaller percentage differences $\Delta_{new}$ may be more intuitive and since both referees prefer $\Delta_{new}$ we have revised the manuscript accordingly.*

- Page 19 Line 10: It may not be valuable to compare secondary inorganic aerosol between MADE and M7. Although M7 only simulate sulfate, the particle is actually (NH4)2SO4, NH4HSO4 in the atmosphere. Many climate model consider sulfate mass as NH4HSO4 instead of SO4. The author did not describe how is sulfate mass treated in M7.

  *The sulfate variable in M7 is interpreted as H2SO4 and not as NH4HSO4. This has been clarified on p5, line 18, as follows:*

"To be consistent with its simplified chemistry scheme, M7 sulfate is interpreted as sulfuric acid. M7 does not account for [nitrogen species and secondary organic aerosols.]"

- Page 28 Line 13: Because the authors are comparing two aerosol modules, it is better to show the figures for both of the two modules. Do they have the same information with the combined data?

*The separate datasets have largely the same information (see Fig. 1 below). However, we do find some regime-dependence, which is most pronounced in the cirrus regime. Here, MADE and M7 feature very different levels of INP. The fit slopes obtained from both dataset together and for the individual dataset vary in the first decimal place. We assume that this variability gives an indication on the systematic error/accuracy associated to the fit slopes. Contrary to this accuracy estimation, the manuscript indicates fit slopes with two decimal places. We have corrected this by rounding the previous values of the fit slopes to one decimal digit in Fig. 14 and Tab. 8. The fit slopes reported in the manuscript are still obtained from the combined data set and not as an average of the two individual datasets. The results of both ways of averaging are practically identical (for the warm phase, combined fitting results in a slope of 0.8, whereas the average of the individual plots gives 0.85 and would thus round to 0.9) but combined fitting seems more accurate and keeps the manuscript shorter.*

- Page 28 Line 16: Change smaller to lower.

*Changed accordingly.*

- Page 31 Line 13: Please clarify these conclusions are probably region-dependent. As I know, the aqueous oxidation of sulfate is sensitive over Europe, but some other regions are not.

*p. 31, line 10 has been changed to:*

"For this case, [a sensitivity study with identical emissions and identical parameterizations of dry and wet deposition for both schemes,] shows [the following sensitivities of simulated atmospheric aerosol burden, sorted in order of decreasing importance:]"

*p. 31, line 19 has been changed to:*

"It needs to be pointed out, however, that the importance of aqueous oxidation displays a strong regional dependence as it depends on cloud cover and droplet pH values."

• Page 31 Line 13: Also, please change chemical reactions to aqueous oxidation, because chemical reactions may mislead readers.

*Changed to:*

"consideration of sulfate production by aqueous oxidation"
* * *
**Fig. 1.** Supplementary plots for Figure 14: M7 (top) vs MADE (bottom) dataset.